# Single-molecule fluorescence microscopy reveals regulatory mechanisms of MYO7A-driven cargo transport in stereocilia of live inner ear hair cells

Takushi Miyoshi [1,2] ✉, Harshad D. Vishwasrao [3], Inna A. Belyantseva [1], Junko Miyoshi [2], Mrudhula Sajeevadathan [2], Yasuko Ishibashi [1,4], Samuel M. Adadey [1], Narinobu Harada [5], Hari Shroff [6] & Thomas B. Friedman [1]

Stereocilia are F-actin-based cylindrical protrusions on the apical surface of inner ear hair cells that function as biological mechanosensors of sound and acceleration. During stereocilia development, specific unconventional myosins transport proteins and phospholipids as cargo and mediate elongation, differentiation and acquisition of the mechanoelectrical transduction (MET). How unconventional myosins localize themselves and cargo in stereocilia using energy from ATP hydrolysis is only partially understood. Here, we developed STELLA-SPIM microscopy to visualize movement of single myosin molecules in live hair cell stereocilia. STELLA-SPIM demonstrated that MYO7A, a component of MET machinery, shows processive movement toward stereocilia tips when chemically dimerized or constitutively activated by missense mutations disabling tail-mediated autoinhibition. Conversely, MYO7A shows step-wise but not processive movement in stereocilia when its tail is tethered to the plasma membrane or F-actin in the presence of MYO7A interacting partners. We posit that MYO7A dimerizes and moves processively in stereocilia when unleashed from autoinhibition.

Hearing loss affects approximately 20% of the population worldwide[1] and is largely classified into sensorineural and conductive hearing loss[2]. Sensorineural hearing loss is usually caused by various factors damaging the inner ear, including genetic variants, noise exposure, ototoxic drugs and aging. In the inner ear, each hair cell develops stereocilia on its apical surface, which are biological mechanosensors of sound or acceleration[3]. Morphologically, stereocilia are cylindrical filamentous actin (F-actin)-based protrusions that are organized in rows of increasing height and equipped with mechanoelectrical transduction (MET) channels at their distal ends except for those in the tallest row. MET channels are gated by stereociliary tip links physically connecting MET channels to a spot on the side of adjacent longer stereocilia[4], namely the upper tip-link density (UTLD) based on its high electron scattering in transmission electron microscopy. When sound

[1]Laboratory of Molecular Genetics, National Institute on Deafness and Other Communication Disorders, National Institutes of Health, Bethesda, MD, USA. [2]Division of Molecular and Integrative Physiology, Department of Biomedical Sciences, Southern Illinois University School of Medicine, Carbondale, IL, USA. [3]Advanced Imaging and Microscopy Resource, National Institute of Biomedical Imaging and Bioengineering, National Institutes of Health, Bethesda, MD, USA. [4]Inner Ear Gene Therapy Program, National Institute on Deafness and Other Communication Disorders, National Institutes of Health, Bethesda, MD, USA. [5]Hearing Research Laboratory, Harada ENT Clinic, Higashi-Osaka, Osaka, Japan. [6]Janelia Research Campus, Howard Hughes Medical Institute, Ashburn, VA, USA. ✉e-mail: tmiyoshi26@siumed.edu

or acceleration deflects stereocilia toward the longer side[5], $K^+$ and $Ca^{2+}$ ions in the endolymph stream into the hair cell cytoplasm through MET channels, depolarize the plasma membrane and finally release glutamate as a neurotransmitter from the basal surface[6–8]. Degeneration of hair cells and stereocilia is a pathological finding that often accompanies sensorineural hearing loss[9]. Recently, *OTOF* gene therapy in deaf humans has successfully restored hearing[10–12]. However, there are still no clinically useful treatments to regenerate hair cells or their stereocilia, which is a reason why sensorineural hearing loss often becomes permanent[13].

Wild-type motor activities of a few unconventional myosins play essential roles when microvilli-like precursors of stereocilia on the apical surface of immature hair cells differentiate into mature stereocilia[14,15]. Unconventional myosins have a conserved motor domain but the tail domain of each myosin is divergent from one another. During stereocilia elongation, thickening and functional maturation, the unique tail domains of different unconventional myosins anchor specific proteins and phospholipids and transport them along the unidirectionally oriented actin filaments of the stereociliary core[16]. For example, MYO7A interacts through its tail with two scaffolding proteins, SANS and USH1C (harmonin), and localizes them to the UTLD of mature stereocilia[17]. UTLDs connect stereociliary tip links, which consist of CDH23 dimers and PCDH15 dimers[18,19], to the F-actin core of the longer stereocilium on the CDH23 side[20]. The PCDH15 side is connected to the MET channel complex, which is composed of TMC1/TMC2 channel proteins and accessory proteins, TMIE, CIB2 and LOXHD1[21–25], at the tip of a shorter stereocilium. Among MYO7A cargos, harmonin b isoform harboring the F-actin binding Proline, Serine and Threonine-rich (PST) domain[26] is essential for forming the UTLD and tethering tip links to the F-actin core[20]. Harmonin b and PCDH15 are absent near stereocilia tips in mice with defective MYO7A function ($Myo7a^{4626SB/4626SB}$)[26–28]. MYO15A elongates F-actin cores by transporting its cargos, such as WHRN (whirlin) and EPS8 (epidermal growth factor receptor kinase substrate 8)[29,30], and also by nucleating actin monomers with its motor domain[31]. MYO3A interacts with ESPN (espin) isoform 1 and ESPNL (espin-like), crucial for the stereocilia staircase architecture[32–34] although localization of these proteins in stereocilia may not completely depend on the motor activities of class-III myosins[35]. To date, variants of *MYO3A*, *MYO6*, *MYO7A* and *MYO15A* are associated with human hereditary nonsyndromic hearing loss[36]. Loss-of-function variants of *MYO7A* also result in autosomal recessive Usher syndrome type 1 characterized by congenital hearing loss, vestibular dysfunction and progressive retinal degeneration[37].

Nevertheless, it is not fully understood how unconventional myosins localize themselves and their cargo in stereocilia using the energy from ATP hydrolysis. It is well known that MYO5A, which is involved in vesicle transport, can spontaneously homo-dimerize using the coiled-coil domain in the tail and then "walk" on F-actin[38]. However, the unconventional myosins crucial for normal hearing are unlikely to spontaneously dimerize. Although MYO6 and MYO7A have predicted coiled-coil domains in the tail, recent studies indicate that these two myosins show cargo-mediated dimerization to walk on F-actin[39–44] and use these "coiled-coil domains" as a single α-helix to extend the neck[45,46]. Class-III myosins have tails lacking apparent dimerization machinery[47]. These myosins are proposed to move like an "inchworm" in stereocilia tethering its tail to F-actin[33,48], but this notion lacks experimental support. It is also uncertain how MYO15A traffics in stereocilia although recombinant MYO15A fragments synthesized in Sf9 cells are partially dimerized by co-purified CETN2 bound to the third IQ motif[49]. Understanding how myosins' motor activities are regulated and utilized in developing stereocilia is key to elucidating the pathophysiology of sensorineural hearing loss and to developing possible therapeutic interventions, for example, to reinstate the development of stereocilia on the apical surface of damaged hair cells and restore MET machinery in degenerating stereocilia.

In this study, we developed a methodology for single-molecule fluorescence microscopy that is applicable to organelles protruding from the apical surface of tissue and visualize myosin molecules at work in stereocilia. To make the focal plane coincident with stereocilia, we employed a dual-view inverted selective plane illumination microscope (diSPIM)[50], which we previously used for multiplexed super-resolution microscopy to detect single molecules of fluorescently labeled imaging probes[51]. Then, we experimentally addressed how MYO7A can transport components of the tip-link complex using a monomeric MYO10 head (i.e., lacking the coiled-coil domain for anti-parallel dimerization[52]) as a positive control in some experiments. We detected processive and directional movement of MYO7A toward stereocilia tips when it is forcefully dimerized or constitutively activated by missense mutations that disable tail-mediated autoinhibition of motor activity[53,54]. Processive movement is not detected when MYO7A is tethered to the plasma membrane or F-actin of stereocilia as mediated by MYO7A interacting partners, such as PCDH15, CDH23[18,19] or harmonin b[20]. These results indicate that MYO7A "walks" in stereocilia as a dimer when unleashed from a tail-mediated autoinhibition. Using these data, we also discuss how myosin motor activities help with the formation of tip-link complexes in developing stereocilia as well as general applications of our STELLA-SPIM (Single-molecule Tracking and Enhanced Localization in Live Aural specimens using Selective Plane Illumination Microscopy) methodology.

## Results

### Single-molecule microscopy workflow in stereocilia of live hair cells

Our STELLA-SPIM workflow was developed using explant cultures of mouse utricles and saccules, hereafter referred to as "vestibular sensory epithelia", harvested on postnatal day (P) 2 to 5 (Fig. 1a). Stereocilia of utricles and saccules are suitable for single-molecule microscopy because they are straight and as long as 10 μm[55] while auditory stereocilia are much shorter and difficult to align with the focal plane of microscopes. Hair cells in the vestibular sensory epithelia are transfected using a Helios® gene gun[56] to co-express HaloTag-fused proteins of interest and enhanced green fluorescent protein (EGFP) or an EGFP-fused protein as a transfection marker. Transfected sensory epithelia are kept in DMEM/F12 culture medium supplemented with 7% fetal bovine serum (37 °C, 5% $CO_2$) for 16–24 h to allow protein expression. HaloTag-fused protein is fluorescently labeled at a low density with JFX554-conjugated HaloTag ligands[57] and visualized using a diSPIM[50] illuminating with a 561-nm laser. In each sensory epithelium explant transfected using plasmid-coated 1.0-μm gold microcarriers, 5–10 hair cells usually express a sufficient amount of HaloTag-fused protein of interest and maintain intact stereocilia architecture.

The concentration of JFX554-conjugated HaloTag ligand is optimized using vestibular hair cells expressing HaloTag-fused human β-actin (HaloTag-actin) (Fig. 1b). Ligand applied at 0.3 nM or higher concentration shows fluorescent labeling of the entire hair cell. Dense labeling occurs at stereocilia tips (Fig. 1b, arrows) and the cuticular plate, indicating that HaloTag-actin shows a similar distribution to EGFP-fused actin[58]. Fluorescent puncta of single HaloTag-actin molecules appear in the cell body at 0.1 nM (Fig. 1b, arrowheads) and in stereocilia around 0.01 nM (Fig. 1b, open arrowheads) although the optimal concentration depends on the expression level of HaloTag-actin. We applied HaloTag ligand below a concentration of 3 nM to avoid unreacted ligands remaining in the tissue. Photoactivatable dyes, such as PA-JF549[59], are not useful for adjusting the labeling density because we found that an uncontrollable proportion of these dyes were in the bright state before being activated by the 405-nm laser and that these dyes are more susceptible to the excitation laser than non-photoactivatable dyes. Single-molecule microscopy is confirmed by calculating the sum of the intensity of each fluorescent punctum and classifying them using a Gaussian Mixture model[60] (Fig. 1c). Among the

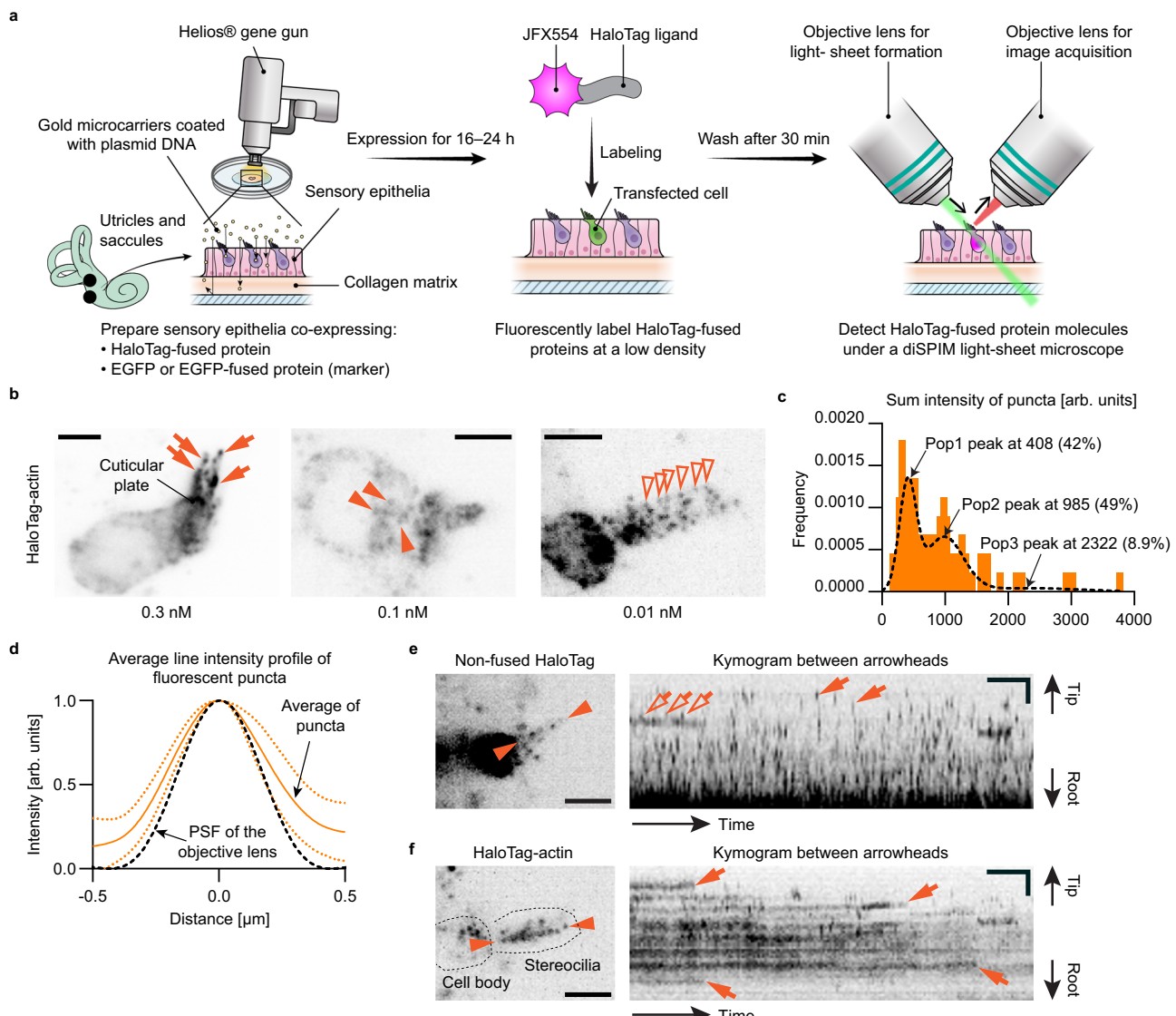

**Fig. 1 | Development of single-molecule microscopy in live hair cells. a** Our workflow for single-molecule microscopy. **b** Labeling density optimization using vestibular hair cells expressing HaloTag-actin. At 0.3 nM JFX554-ligand, the entire cells are labeled, with dense signal at stereocilia tips (arrows) and in the cuticular plate. Fluorescent puncta appear in the cell body at 0.1 nM (arrowheads) and in stereocilia around 0.01 nM (open arrowheads). Maximum projections of volume scans. Exposure, 100 ms per plane at 0.2 kW/cm². Bars, 5 μm. **c** Classification of fluorescent puncta using the Gaussian Mixture model[60]. The sum intensity is calculated by integrating the pixel values after background subtraction. Among the three populations (Pop1, Pop2 and Pop3), the peak intensity of Pop2 (985) is approximately twice that of Pop1 (408), indicating that Pop1 and Pop2 are emitted from one and two fluorophores, respectively. A total of 76 puncta were analyzed from six cells (including **b**, 0.01 nM), using average projections of 12 planes per volume scan. **d** Comparison between the average line intensity profile of

fluorescent puncta (orange solid line) and the theoretical point spread function (PSF) of the objective lens calculated using the Born & Wolf 3D Optical Model[117,118] (black dashed line). The similarity between these two intensity curves suggests that these puncta are emitted from a point source. Fluorescence intensity is an average of 10 puncta in **b** (0.01 nM, Pop1 only). SD, orange dotted lines. **e, f** Representative kymograms of non-fused HaloTag (**e**) and HaloTag-actin (**f**) labeled with 0.1 nM and 0.01 nM JFX554-ligands, respectively. Single-plane images are acquired every 1 s for comparison with MYO7A movement. Kymograms are generated from the line scans between arrowheads. Most non-fused HaloTag molecules disappear after one frame (**e**, arrows) due to diffusion, except for a few molecules (**e**, open arrows). Most HaloTag-actin molecules stay in the same place and disappear due to photobleaching or transition to the dark state (**f**, arrows). Imaging conditions are similar to (**b**). Bars, 5 μm (cell images); 2 μm and 20 s (kymograms). Source data are provided as a Source Data file.

three populations detected (Pop1, Pop2 and Pop3), more than 90% of fluorescent puncta are explained by Pop1 and Pop2 (42% and 49%, respectively). The peak intensity of Pop2 (985) is approximately twice as large as that of Pop1 (408) indicating that Pop1 and Pop2 originate from one and two fluorophores, respectively. The intensity profile along a line is also consistent with the point spread function of the objective lens (Fig. 1d) suggesting that each fluorescent punctum is emitted from a point source.

Time-lapse images of non-fused HaloTag and HaloTag-actin molecules are acquired to evaluate how proteins are visualized when they diffuse and stably bind to the F-actin core, respectively (Fig. 1e, f). Single-plane time-lapse images are acquired every 1 s (s) and compared with subsequent MYO7A images. Consistent with diffusion, most non-fused HaloTag molecules stay in the same position no longer than one frame (Fig. 1e, arrows; Supplementary Movie 1) except for a few molecules that remain in one place probably due to non-specific binding to the stereocilium structure (Fig. 1e, open arrows). This finding is also supported by the short half-life ($T_{1/2}$) of non-fused HaloTag molecules dwelling in stereocilia ($T_{1/2}$ = 0.46 frames; 95% confidence interval [CI] = 0.45–0.47; Supplementary Fig. 1a). In

contrast, most HaloTag-actin molecules stay in one place and show trajectories parallel to the time axis (X-axis) of kymograms (Fig. 1f, arrows; Supplementary Movie 2). In live hair cells, regression of HaloTag-actin molecules is approximately 30-fold slower ($P < 0.001$) than non-fused HaloTag ($T_{1/2} = 13.9$ frames; 95% CI = 13.3–14.5; Supplementary Fig. 1b) indicating the presence of HaloTag-actin molecules stably bound to the F-actin core. Interestingly, regression of HaloTag-actin molecules occurs more slowly ($P < 0.001$) in fixed cells ($T_{1/2} = 107$ frames; 95% CI = 69–231; Supplementary Fig. 1b) at a rate close to the photobleaching (hereafter meaning dye decay and transition to the dark state[61]) rate of JFX554 dyes ($T_{1/2} = 61$ frames; 95% CI = 52–72; Supplementary Fig. 1c). HaloTag-actin molecules in live cells may photobleach more rapidly or dissociate from actin filaments more easily than in fixed cells. Alternatively, some HaloTag-actin molecules may transiently bind to the F-actin core in live cells[62]. Kymograms show sudden disappearance of HaloTag-actin molecules consistent with the quantum behavior of single fluorophores (Fig. 1f, arrows). The quantum behaviors of dyes are more clearly visualized in control fixed cells acquired every 100 milliseconds (ms) including dyes recovering from the dark state to the bright state (Supplementary Fig. 1d).

## MYO7A dimers directionally moving in stereocilia

We developed imaging conditions to detect MYO7A trafficking in stereocilia using the heavy meromyosin-like fragment of mouse MYO7A (MYO7A-HMM; Fig. 2a). MYO7A-HMM has a domain composition similar to that of heavy meromyosin (HMM), a protein fragment of myosin II obtained by trypsinization[44] consisting of a motor domain and a neck necessary for a power stroke on F-actin[63,64]. MYO7A-HMM can walk processively as a dimer in F-actin protrusions such as filopodia and microvilli[44,65]. In this study, we expressed MYO7A-HMM with a HaloTag at the N-terminus for fluorescent labeling and utilized the p.F36V substitution mutant of FK506 binding protein 12 (FKBP) at the C-terminus for conditional dimerization under treatment with an FK506-derived bivalent ligand, AP20187[66] (HaloTag-MYO7A-HMM-FKBP; Fig. 2b). Conditional dimerization is useful to initiate HaloTag-MYO7A-HMM-FKBP trafficking only when cells are ready for imaging. Confocal microscopy shows that HaloTag-MYO7A-HMM-FKBP expressed in vestibular hair cells forms large protein blobs at stereocilia tips under AP20187 treatment (Fig. 2c, left panels). This result indicates that MYO7A-HMM dimers move toward the barbed ends of unidirectional F-actin bundles of stereocilia. Without AP20187 treatment, HaloTag-MYO7A-HMM-FKBP does not accumulate at stereocilia tips (Fig. 2c, right panels).

Single HaloTag-MYO7A-HMM-FKBP molecules are successfully detected within stereocilia using 0.3–0.6 nM JFX554 ligands (Fig. 2d; Supplementary Movie 3). A higher concentration of JFX554 ligands is required for HaloTag-MYO7A-HMM-FKBP than for HaloTag-actin, as MYO7A expression level is lower than that of β-actin. Time-lapse images after AP20187 treatment show HaloTag-MYO7A-HMM-FKBP molecules moving directionally toward stereocilia tips (Fig. 2d, magenta circles). Kymograms show continuous trajectories consistent with processive "walking" of HaloTag-MYO7A-HMM-FKBP dimers (Fig. 2d, arrows). The velocity of movement varies between dimers (Supplementary Fig. 2a, b; representative kymograms) although there are no distinct populations of slow and rapid movements (Fig. 2e). The average velocity is $101 \pm 53$ nm/s ($n = 42$; mean ± standard deviation [SD]), which is 10-fold faster than the movement of human recombinant MYO7A-HMM fused with a leucine zipper dimerization sequence on permeabilized filopodia ($9.5 \pm 0.4$ nm/s; mean ± standard error [SE])[67]. This difference can be partially attributed to the temperature (37 °C in our study vs. 25 °C in the previous study) although MYO10 shows only a 2-fold increase in velocity in live-cell filopodia ($578 \pm 174$ nm/s at 25 °C vs. $840 \pm 210$ nm/s at 37 °C; mean ± SD)[68]. Run lengths are $2.3 \pm 1.0$ μm (Fig. 2f, $n = 42$) and longer than the previous

in vitro study using single actin filaments ($0.71 \pm 0.09$ μm; mean ± SE)[67]. In stereocilia of live hair cells, MYO7A-HMM dimers may be allowed to stably bind to F-actin because protein diffusion is restricted by the plasma membrane[69]. HaloTag-MYO7A-HMM-FKBP does not show processive movement without AP20187 (Fig. 2g; Supplementary Movie 4). However, it is notable that some MYO7A-HMM-FKBP monomers may relocate in stereocilia (Supplementary Movie 4, circles) and appear as stepwise trajectories in kymograms (Fig. 2g, open arrows in the inset). The frequency of processive movement is roughly compared across cells, assuming that each transfected hair cell expresses an almost equal amount of HaloTag-fused myosin (hereafter referred to as "semi-quantification"). Processive movement is detected at a significantly higher frequency in AP20187-treated cells than in non-treated cells (Supplementary Fig. 2c, $P = 0.0004$).

## Processive movement of constitutively active MYO7A mutants in stereocilia

The imaging condition for MYO7A-HMM dimers was used to test the hypothesis that MYO7A traffics as dimers or oligomers in stereocilia (Fig. 3). However, HaloTag-fused full-length (wild-type) MYO7A does not show directional movement at a detectable frequency ($n = 10$ cells, image not shown). We considered the possibility that full-length MYO7A takes a backfolded autoinhibitory conformation[53,54], and designed two HaloTag-fused constitutively active MYO7A mutants, HaloTag-MYO7A-RK/AA and HaloTag-MYO7A-ΔSH3-ΔM/F2 (Fig. 3a). MYO7A-RK/AA has two missense mutations in the "RGSK" motif of the second MyTH4-FERM (M/F2) domain, which is essential for tail-mediated autoinhibition, and changes this sequence to "AGSA"[70]. MYO7A-ΔSH3-ΔM/F2 truncates the SH3 and M/F2 domains. Confocal microscopy shows that HaloTag-MYO7A-RK/AA often accumulates at stereocilia tips, while HaloTag-fused full-length MYO7A diffusely distributes in stereocilia (Fig. 3b, representative images shown). HaloTag-MYO7A-ΔSH3-ΔM/F2 distributes diffusely in stereocilia but seldomly accumulates at stereocilia tips (Fig. 3c). Pairwise comparisons using Fisher's exact test show $P = 0.042$ (nominally significant) between full-length MYO7A and MYO7A-RK/AA on the formation of protein blobs at stereocilia tips (Supplementary Fig. 3a). It is also notable that HaloTag-fused mouse MYO7A used in this study localizes differently from endogenous MYO7A in guinea pig and rat inner ear, which concentrates in the UTLD[17]. To examine the possibility that a HaloTag stabilizes the autoinhibitory conformation of MYO7A and accounts for this discrepancy, we expressed full-length mouse MYO7A in vestibular hair cells, fusing a small HA tag either at the N- or C-terminus (Supplementary Fig. 3b). Immunostaining revealed that HA-tagged MYO7A is diffusely distributed along stereocilia, similar to HaloTag-fused MYO7A, suggesting that ectopically expressed MYO7A remains autoinhibited regardless of tag type or position. Additional factors are likely required to activate ectopically expressed MYO7A in hair cells.

Under single-molecule microscopy, a few HaloTag-MYO7A-RK/AA molecules move toward stereocilia tips (Fig. 3d; Supplementary Movie 5). Movement of MYO7A-RK/AA is directional and processive, resembling the movement of MYO7A-HMM dimers. This result suggests that MYO7A can dimerize on the F-actin cores of stereocilia when the motor domain is exposed, for example, by cargo bound to the tail[53,71,72]. Similar processive movement is observed in cells expressing HaloTag-MYO7A-ΔSH3-ΔM/F2 (Fig. 3e; Supplementary Movie 6). MYO7A-RK/AA molecules show processive movement more frequently than full-length MYO7A and MYO7A-ΔSH3-ΔM/F2 (Supplementary Fig. 3c, $P = 0.0010$ and 0.0028), which is consistent with the frequent accumulation of MYO7A-RK/AA at stereocilia tips compared with full-length MYO7A and MYO7A-ΔSH3-ΔM/F2 (Supplementary Fig. 3a). The velocity during directional movement is not significantly different among MYO7A-RK/AA, MYO7A-ΔSH3-ΔM/F2 and MYO7A-HMM dimers (Supplementary Fig. 3d). MYO7A-RK/AA and MYO7A-ΔSH3-ΔM/F2 show statistically shorter run lengths than MYO7A-HMM

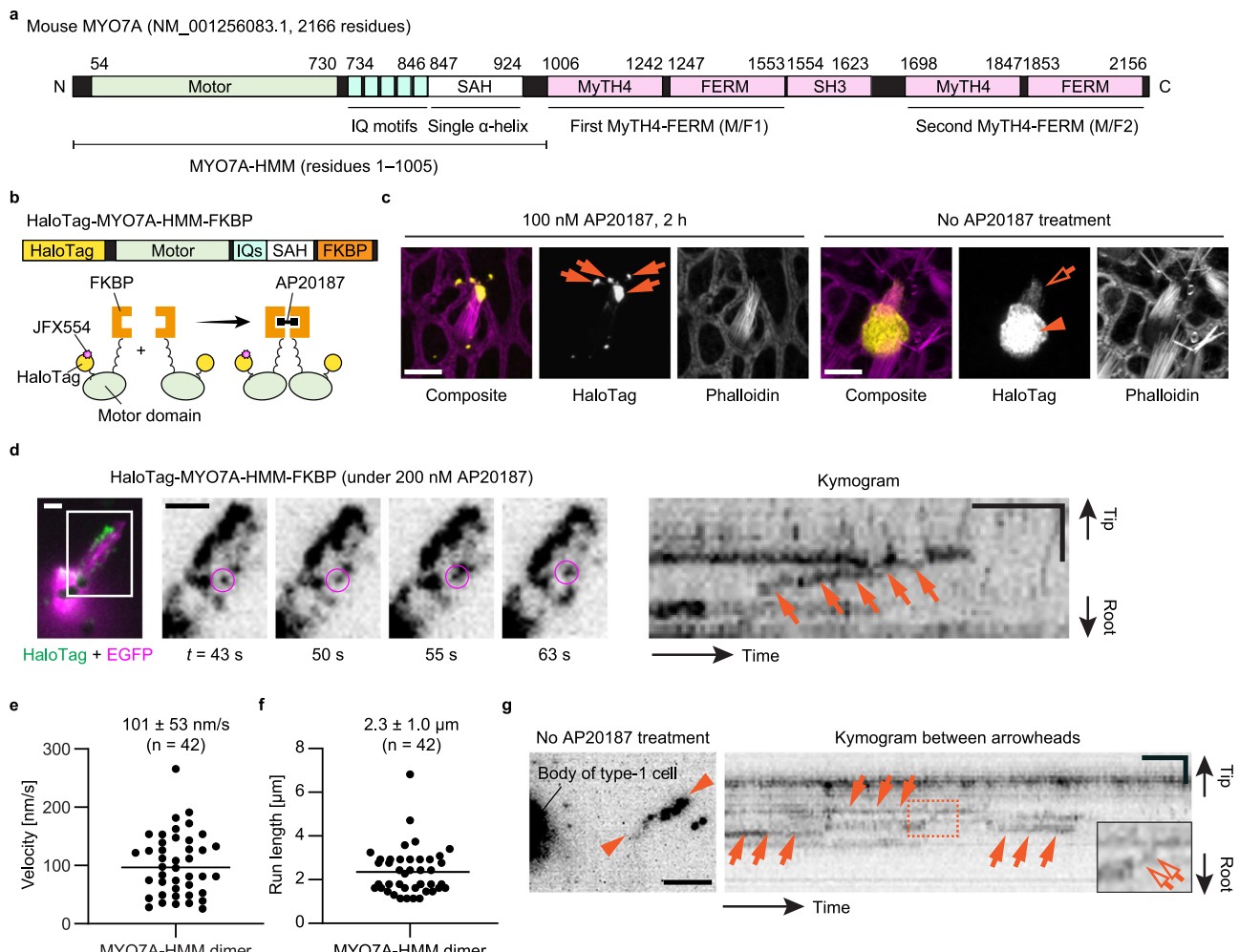

**Fig. 2 | MYO7A-HMM dimers directionally moving in stereocilia. a** Domain structures of mouse MYO7A (NM_001256083.1) and its heavy-meromyosin-like fragment (MYO7A-HMM) used in this study. **b** Schematic diagrams showing the structure of HaloTag-MYO7A-HMM-FKBP and conditional dimerization under AP20187 treatment. Note that only a small portion of HaloTag is labeled in our single-molecule microscopy. **c** Confocal microscopy showing AP20187-dependent accumulation of HaloTag-MYO7A-HMM-FKBP at stereocilia tips. Vestibular hair cells (P2) expressing HaloTag-MYO7A-HMM-FKBP are incubated with or without 100 nM AP20187 for 2 h. Samples are fixed and stained with 200 nM JFX554-conjugated HaloTag ligands (yellow) and Alexa405-phalloidin (magenta). HaloTag-MYO7A-HMM-FKBP localizes to large protein blobs at stereocilia tips in AP20187-treated cells (arrows), and diffusely in the cuticular plate (arrowhead) and stereocilia (open arrow) in non-treated cells. Bars, 5 μm. **d** Directional and processive movement of MYO7A-HMM dimers in stereocilia under 200 nM AP20187

treatment. Time-lapse images (grayscale) are shown for the rectangular region in the pseudo-colored image (HaloTag in green and EGFP in magenta). A moving molecule is indicated by magenta circles in time-lapse images and as a continuous trajectory in the kymogram (arrows). JFX554, 0.3 nM. Single-plane time-lapse, every 1 s. Bars, 5 μm (time-lapse images); 2 μm and 20 s (kymogram). **e, f** Velocity and run lengths of MYO7A-HMM dimers directionally moving in stereocilia. The means ± SDs are 101 ± 53 nm/s and 2.3 ± 1.0 μm (*n* = 42). **g** Behavior of HaloTag-MYO7A-HMM-FKBP molecules in non-treated cells. Trajectories in the kymogram (arrows) are parallel to the time axis and consistent with molecules staying at the same position. Stepwise movement toward the stereocilia tips is occasionally detected (open arrows in the inset, magnifying the rectangle). Imaging conditions and scale bars are the same as in grey-scale images in (**d**). Source data are provided as a Source Data file.

dimers (Supplementary Fig. 3e, *P* = 0.0087 and 0.037) although this small difference may not be biologically important. Processive movement of MYO7A-ΔSH3-ΔM/F2 indicate that MYO7A can dimerize (or oligomerize) using motifs in the neck or the first MyTH4-FERM (M/F1) domain. The SH3 and/or M/F2 domains may be necessary for MYO7A to traffic efficiently in stereocilia. For example, harmonin can interact with the M/F2 domain and expose the motor domain, which can lead to dimerization of MYO7A[73].

### Membrane-anchored MYO7A does not show processive movement

In addition to dimerization, we tested how MYO7A can move in stereocilia when anchored to the plasma membrane (Fig. 4) because MYO7A in the UTLD interacts with CDH23 via SANS and harmonin and is likely anchored to the plasma membrane[26,74,75]. PCDH15 may also anchor

MYO7A to the plasma membrane because the CD2 isoform of PCDH15 has an intracellular region that can interact with the SH3 domain of MYO7A[28]. Here, we used a single-pass transmembrane motif of the human Interleukin 2 receptor alpha chain (IL2Rα)[76] as a membrane anchor (Fig. 4a). IL2Rα and MYO7A-HMM are conditionally heterodimerized using FKBP and FKBP-Rapamycin binding protein (FRB) fused to the C-terminus of these proteins, respectively (IL2Rα-EGFP-FKBP and HaloTag-MYO7A-HMM-FRB in Fig. 4a) and AP21987, a rapamycin analog, added to the culture medium[77]. EGFP is inserted between IL2Rα and FKBP as a transfection marker. For positive control, a monomeric MYO10 head (MYO10-MD) is constructed from bovine MYO10 eliminating the entire tail and the neck coiled-coil domain for anti-parallel dimerization because this MYO10 fragment can traffic in filopodia using the plasma membrane as a scaffold[52]. Confocal microscopy shows that HaloTag-MYO7A-HMM-FRB successfully distributes

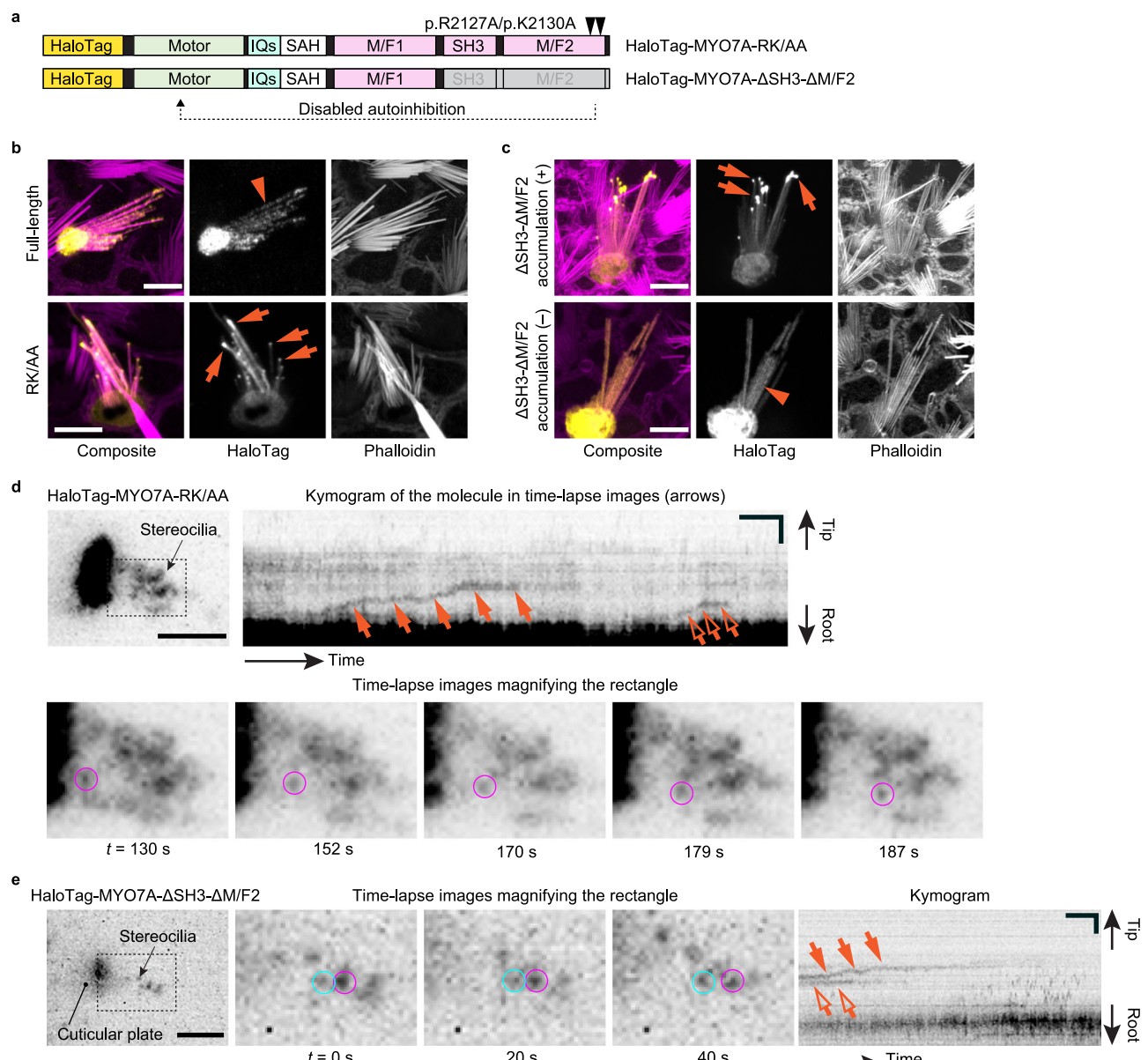

**Fig. 3 | Directional movement of constitutively active MYO7A mutants.**
**a** Structures of two constitutively active HaloTag-fused MYO7A mutants. HaloTag-MYO7A-RK/AA has two missense mutations, p.R2127A and p.K2130A, that we insert in the second MyTH4-FERM (M/F2) domain, referring to the study using human MYO7A[70]. HaloTag-MYO7A-ΔSH3-ΔM/F2 has a truncated tail. These mutations are introduced to remove the tail-mediated autoinhibition of the motor domain.
**b** Representative confocal images of HaloTag-fused full-length MYO7A and Halo-Tag-MYO7A-RK/AA expressed in vestibular hair cells (P2–5). Full-length MYO7A diffusely distributes in stereocilia (arrowhead) while HaloTag-MYO7A-RK/AA accumulates at stereocilia tips (arrows) of some cells, indicating directional movement toward stereocilia tips. Bars, 5 μm. **c** Confocal images of MYO7A-ΔSH3-ΔM/F2 expressed in vestibular hair cells (P2–5). This mutant accumulates at

stereocilia tips in a small number of cells (arrows) and localizes diffusely in stereocilia in other cells (arrowhead). Bars, 5 μm. **d** Single-molecule microscopy of HaloTag-MYO7A-RK/AA. Time-lapse images (lower panels) show a molecule directionally moving in stereocilia (magenta circles). The kymogram (upper right panel) illustrates the processive movement of this molecule (arrows) and another molecule (open arrows). JFX554, 0.3 nM. Single-plane time-lapse, every 1 s. Bars, 5 μm (time-lapse images); 2 μm and 20 s (kymogram). **e** Single-molecule microscopy of HaloTag-MYO7A-ΔSH3-ΔM/F2. Time-lapse images and the kymogram show the processive and directional movement of a molecule (magenta circles and arrows). A stationary molecule is indicated for comparison (cyan circles and open arrows). Imaging conditions and scale bars are the same as (**d**).

near the plasma membrane after AP21987 treatment (Fig. 4b). The positive control, HaloTag-MYO10-MD-FRB, accumulates weakly at stereocilia tips in a few cells without AP21987 (Fig. 4c, arrowhead) and densely at stereocilia tips after AP21987 treatment (Fig. 4c, arrows). Excess IL2Rα-EGFP-FKBP sometimes accumulates in vesicles without apparent damage to stereocilia (Fig. 4b, open arrowheads).

Single-molecule microscopy shows that MYO7A-HMM anchored to the plasma membrane does not show processive movement in stereocilia (Fig. 4d; Supplementary Movie 7). Instead, we found that

membrane-anchored MYO7A-HMM sometimes shows stepwise directional movement (Fig. 4d, arrows and open arrows). However, stepwise movement may not be solely due to anchoring to the membrane because MYO7A-HMM monomers show similar movement at a low frequency (Fig. 2g). Membrane-anchored MYO7A-HMM shows stepwise movement at a higher frequency than unanchored MYO7A-HMM, but this difference is not statistically significant (Supplementary Fig. 4a). In contrast, MYO10-MD markedly changes its movement upon membrane anchoring (Fig. 4e; Supplementary Movies 8 and 9),

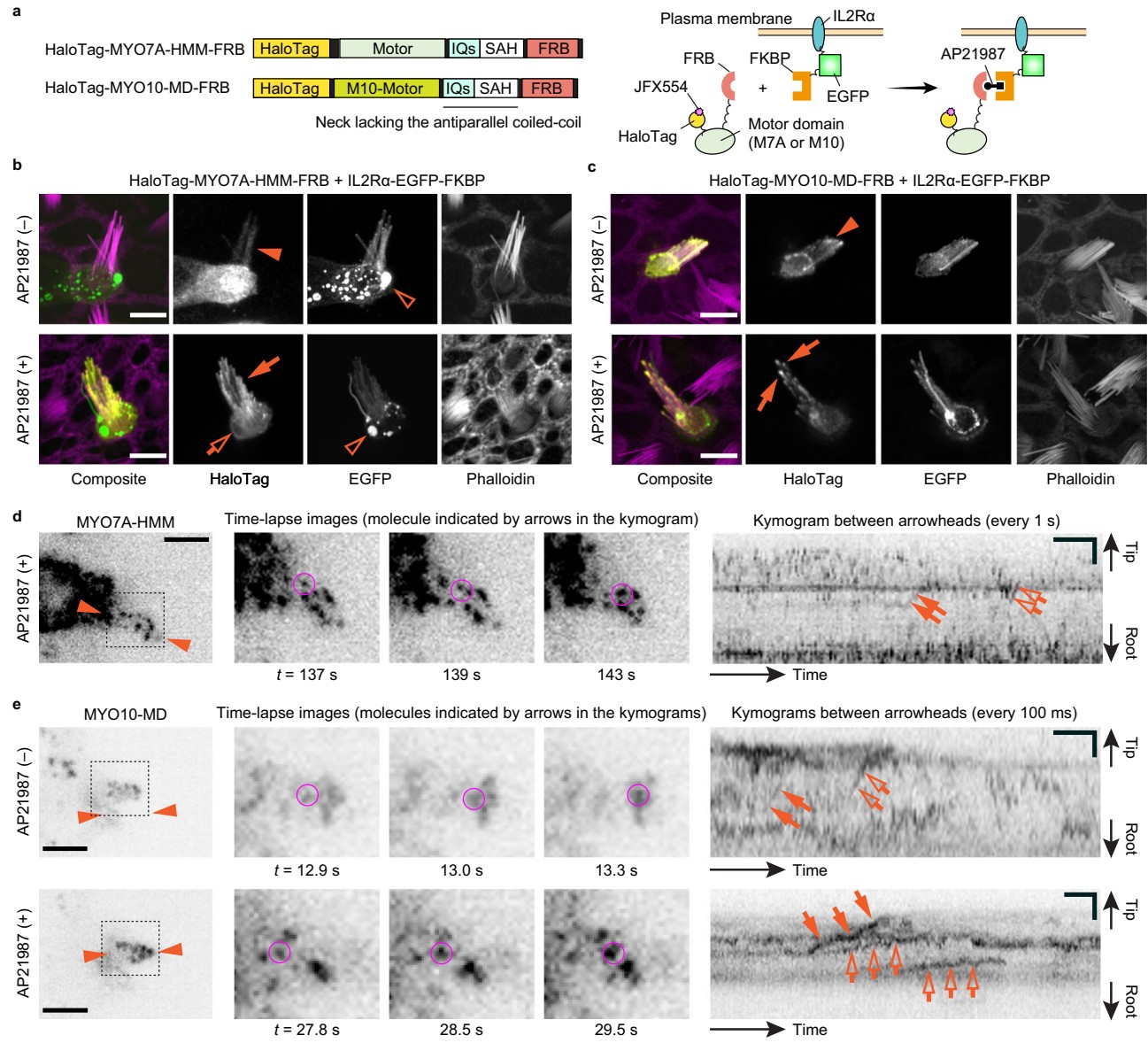

**Fig. 4 | Movement of membrane-anchored MYO7A-HMM and MYO10-MD.**
**a** Domain structures of HaloTag-MYO7A-HMM-FRB and HaloTag-MYO10-MD-FRB and a schematic diagram showing membrane anchoring by IL2Rα-EGFP-FKBP (human Interleukin 2 receptor alpha chain fused with EGFP and FKBP). Membrane anchoring is mediated by conditional heterodimerization between FRB and FKBP under AP21987 treatment. MYO10-MD is a control monomeric myosin motor head[52]. **b** Confocal images showing AP21987-dependent membrane anchoring of HaloTag-MYO7A-HMM-FRB. Vestibular hair cells (P2) co-expressing HaloTag-MYO7A-HMM-FRB and IL2Rα-EGFP-FKBP are incubated with or without 500 nM AP21987 for 2 h. HaloTag-MYO7A-HMM-FRB distributes diffusely along stereocilia of untreated cells (arrowhead) but accumulates along the plasma membrane of stereocilia (arrow) and at the edge of the cuticular plate (open arrow) in AP21987-treated cells. Excess IL2Rα-EGFP-FKBP sometimes accumulates in vesicles without apparent damage to stereocilia (open arrowheads). Bars, 5 μm. **c** Confocal images showing AP21987-dependent membrane anchoring of MYO10-MD and its localization changes. HaloTag-MYO10-MD-FRB accumulates at stereocilia tips weakly in a few untreated cells (arrowhead), suggesting that a small number of MYO10-MD molecules can move directionally without a scaffold. Increased accumulation of HaloTag-MYO10-MD-FRB at stereocilia tips in AP21987-treated cells (arrows) indicates directional movement enhanced by membrane anchoring. This localization change in stereocilia is consistent with a previous study of MYO10-MD in filopodia[52]. Bars, 5 μm. **d** Single-molecule microscopy of membrane-anchored MYO7A-HMM. Staircase-like trajectories in the kymogram indicate stepwise movement of MYO7A-HMM (arrows and open arrows). The molecule indicated by arrows is shown in time-lapse images (magenta circles). AP21987, 500 nM. Single-plane time-lapse, every 1 s. Bars, 5 μm (cell image); 20 s and 2 μm (kymogram). **e** Single-molecule microscopy of unanchored and membrane-anchored MYO10-MD. Before AP21987 treatment, a small number of MYO10-MD molecules show rapid directional and processive movement toward stereocilia tips (arrows and open arrows in the upper kymogram). After AP21987 treatment, MYO10-MD molecules show slower directional and processive movement (arrows and open arrows in the lower kymogram). Molecules indicated by arrows are shown in time-lapse images (magenta circles). AP21987, 500 nM. Single-plane images, every 100 ms. Bars, 5 μm (cell images); 2 s and 2 μm (kymograms).

indicating that IL2Rα-EGFP-FKBP does function as a scaffold for some myosins to traffic in stereocilia. Consistent with its weak accumulation at stereocilia tips (Fig. 4c, arrowhead) and filopodia[52], MYO10-MD can show rapid directional and processive movement toward stereocilia tips without anchoring to the membrane (Fig. 4e, arrows and open arrows in the upper panel). After AP21987 treatment, MYO10-MD begins to show slower processive movement toward stereocilia tips (Fig. 4e, arrows and open arrows in the lower panel, additional

kymograms in Supplementary Fig. 4c) and sometimes processive retrograde movement (Supplementary Fig. 4c, arrowheads). Although the frequency of processive movement does not change significantly by membrane anchoring (Supplementary Fig. 4b), the velocity of processive movement is significantly different between MYO10-MD monomers (1800 ± 490 nm/s, $n = 14$) and membrane-anchored MYO10-MD (701 ± 297 nm/s, $n = 32$; Supplementary Fig. 4d, $P < 0.0001$). Membrane-anchored MYO7A-HMM moves at 88 ± 27 nm/s ($n = 7$), which is not significantly different from MYO7A-HMM dimers (101 ± 53 nm/s) (Supplementary Fig. 4d) but much slower than MYO10-MD (Supplementary Fig. 4d, $P < 0.0001$ vs. MYO10-MD monomers and vs. membrane-anchored MYO10-MD). Run lengths are slightly shorter for membrane-anchored MYO7A-HMM compared with membrane-anchored MYO10-MD ($P = 0.018$) and MYO7A-HMM dimers ($P = 0.011$) Supplementary Fig. 4e).

### Possible stepwise movement of MYO7A coupled with a harmonin b fragment

Finally, we tested the hypothesis that MYO7A can use harmonin b to move in a stereocilium because harmonin can tether the MYO7A tail to F-actin (Fig. 5). To date, three classes of harmonin isoforms, a, b and c, have been identified in hair cells[78]. While all isoforms have the second PDZ domain to interact with MYO7A via SANS[79], only harmonin b can bind to F-actin using the Proline, Serine and Threonine-rich (PST) domain[26] (Fig. 5a). The PST domain of harmonin b is necessary for tethering tip links to F-actin as indicated by *dfcr* mice, which have a 12.8 kb deletion including the exons for the PST domain[20]. Mice with homozygous loss of this region develop stereocilia lacking the UTLD and exhibit profound hearing loss[20]. However, morphologically normal tip links in *dfcr* mice[20] suggest that amino-acid residues removed by this deletion (hereafter, the DFCR fragment) are not necessary for tip-link formation. Behaviors of MYO7A-HMM coupled with the DFCR fragment may explain how tip links maintain their position and tension on mature stereocilia.

To evaluate how the DFCR fragment affects MYO7A movement in stereocilia, we conditionally tethered the C-terminus of MYO7A-HMM or MYO10-MD to this fragment. MYO7A-HMM and MYO10-MD are expressed with a HaloTag at the N-terminus and FKBP at the C-terminus (HaloTag-MYO7A-HMM-FKBP and HaloTag-MYO10-MD-FKBP in Fig. 5b). The DFCR fragment (residues 296–728 of NM_01163733) is fused with FRB at the N-terminus and EGFP at the C-terminus as a transfection marker (FRB-DFCR-EGFP in Fig. 5a). Compared with the amino acid sequence deleted in *dfcr* mice, our DFCR fragment omits the N-terminal two residues that belong to the second PDZ domain. Before AP21987 treatment, confocal microscopy shows that HaloTag-MYO7A-HMM-FKBP diffusely distributes in stereocilia (Fig. 5c, orange arrowhead). HaloTag-MYO10-MD-FKBP shows a similar distribution (Fig. 5d, orange arrowhead) but weakly accumulates at stereocilia tips in some cells (image not shown). FRB-DFCR-EGFP localizes diffusely in stereocilia and at the edge of the cuticular plate (Fig. 5c, d, white arrowheads and arrows), suggesting its interaction with F-actin. Interestingly, FRB-DFCR-EGFP shows accumulation at or near stereocilia tips (Fig. 5c, d, orange open arrowheads), which may co-localize with endogenous harmonin as antiparallel dimers via the second coiled-coil (CC2) domain in our fragment[80]. After AP21987 treatment, HaloTag-MYO7A-HMM-FKBP co-localizes with FRB-DFCR-EGFP at or near stereocilia tips (Fig. 5c, orange arrows) indicating that AP21987 couples MYO7A-HMM with DFCR fragments. Part of HaloTag-MYO7A-HMM-FKBP may be able to traffic toward stereocilia tips using the DFCR fragment as a scaffold since HaloTag-MYO7A-HMM-FKBP sometimes localizes to stereocilia tips without dense FRB-DFCR-EGFP puncta (Fig. 5c, yellow arrows). HaloTag-MYO10-MD-FKBP forms protein blobs at stereocilia tips with FRB-DFCR-EGFP after AP21987 treatment (Fig. 5d, orange arrows and open arrows) suggesting that MYO10-MD molecules can move toward stereocilia tips using DFCR

fragments as a scaffold presumably to tether the MYO10-MD C-terminus to F-actin.

Only stepwise movement are detected for MYO7A-HMM coupled with the DFCR fragment (Fig. 5e, arrows; Supplementary Movie 10). Different from the membrane anchor IL2Rα-EGFP-FKBP, the DFCR fragment may partly function as a scaffold for MYO7A-HMM to move in stereocilia because the frequency of directional movement increases slightly after AP21987 treatment (Supplementary Fig. 5a, $P = 0.033$). This finding is consistent with the AP21987-dependent accumulation of HaloTag-MYO7A-HMM-FKBP without FRB-DFCR-EGFP puncta (Fig. 5c, yellow arrows). The CC1 and CC2 domains in the DFCR fragment are unlikely to dimerize MYO7A-HMM since no processive movement is detected. As shown for IL2Rα-EGFP-FKBP, movement of MYO10 slows down when coupled with the DFCR fragment (Fig. 5f; Supplementary Movie 11) but without changing the frequency of moving molecules (Supplementary Fig. 5b). MYO10-MD moves at 831 ± 456 nm/s ($n = 14$) in cells treated with AP21987, which is significantly slower than 1780 ± 487 nm/s ($n = 14$) in untreated cells (Supplementary Fig. 5c, $P < 0.0001$), suggesting that FRB-DFCR-EGFP can function as a scaffold for MYO10-MD to move in stereocilia. The velocity is not significantly different between the processive movement of MYO7A-HMM dimers (101 ± 53 nm/s) and stepwise movement of MYO7A-HMM coupled with the DRCR fragment (84 ± 44 nm/s, $n = 12$). MYO7A-HMM moves much slower than MYO10-MD (Supplementary Fig. 5c). Run lengths of MYO7A-HMM dimers are slightly longer than MYO7A-HMM coupled with the DFCR fragment, uncoupled MYO10-MD and MYO10-MD coupled with the DFCR fragment (Supplementary Fig. 5d, $P = 0.006$, 0.0019 and 0.0024, respectively). Along with the absent processive movement of membrane-anchored MYO7A-HMM, these results suggest that processive movement of constitutively active mutants used in this study, MYO7A-RK/AA and MYO7A-ΔSH3-ΔM/F2, is driven by their walk as a dimer or an oligomer on the unidirectionally bundled actin filaments in stereocilia.

## Discussion

Stereocilia are intricate mechanosensors composed of more than 500 proteins, including actin monomers, actin regulatory proteins, unconventional myosins and components of the MET channel and tip-link complexes[81]. Among these proteins, unconventional myosins play crucial roles in developing stereocilia by transporting and/or anchoring specific cargo. Nevertheless, it is unclear (1) how these myosins and cargo traffic on the F-actin core of stereocilia, including whether each myosin functions as a monomer, a dimer or an oligomer, and (2) what regulates their motor activities, including the possibility of cargo-mediated dimerization (oligomerization). In this study, we approached these questions through real-time functional analyses based on single-molecule fluorescence microscopy. The mechanisms of myosin-driven active cargo transport in hair cells are important to elucidate the pathophysiology of hereditary hearing loss and develop therapeutic strategies to restore normal hearing.

Processive movement of constitutively active MYO7A mutants in stereocilia can be a good starting point to elucidate how MYO7A motor activity is utilized in developing hair cells (Fig. 6a). In guinea pig and rat inner ears, MYO7A localizes to the UTLD with at least two scaffolding proteins, SANS and harmonin[17–20]. Harmonin b is essential for forming the UTLD and connecting stereociliary tip links to the F-actin core on the CDH23 side although morphologically normal tip links are formed in mice lacking this harmonin isoform[20,78]. MYO7A and tip-link components form an interaction network (Fig. 6b) that may be more complex, including twinfilin-2[82] and PDZD7[83,84]. Movement of MYO7A-RK/AA and MYO7A-ΔSH3-ΔM/F2 (Fig. 3) resembles that of MYO7A-HMM dimers (Fig. 2) but is different from stepwise movement of MYO7A-HMM anchored to the plasma membrane or coupled with a harmonin b fragment (Figs. 4 and 5). Considering that full-length MYO7A does not show processive movement in stereocilia, MYO7A

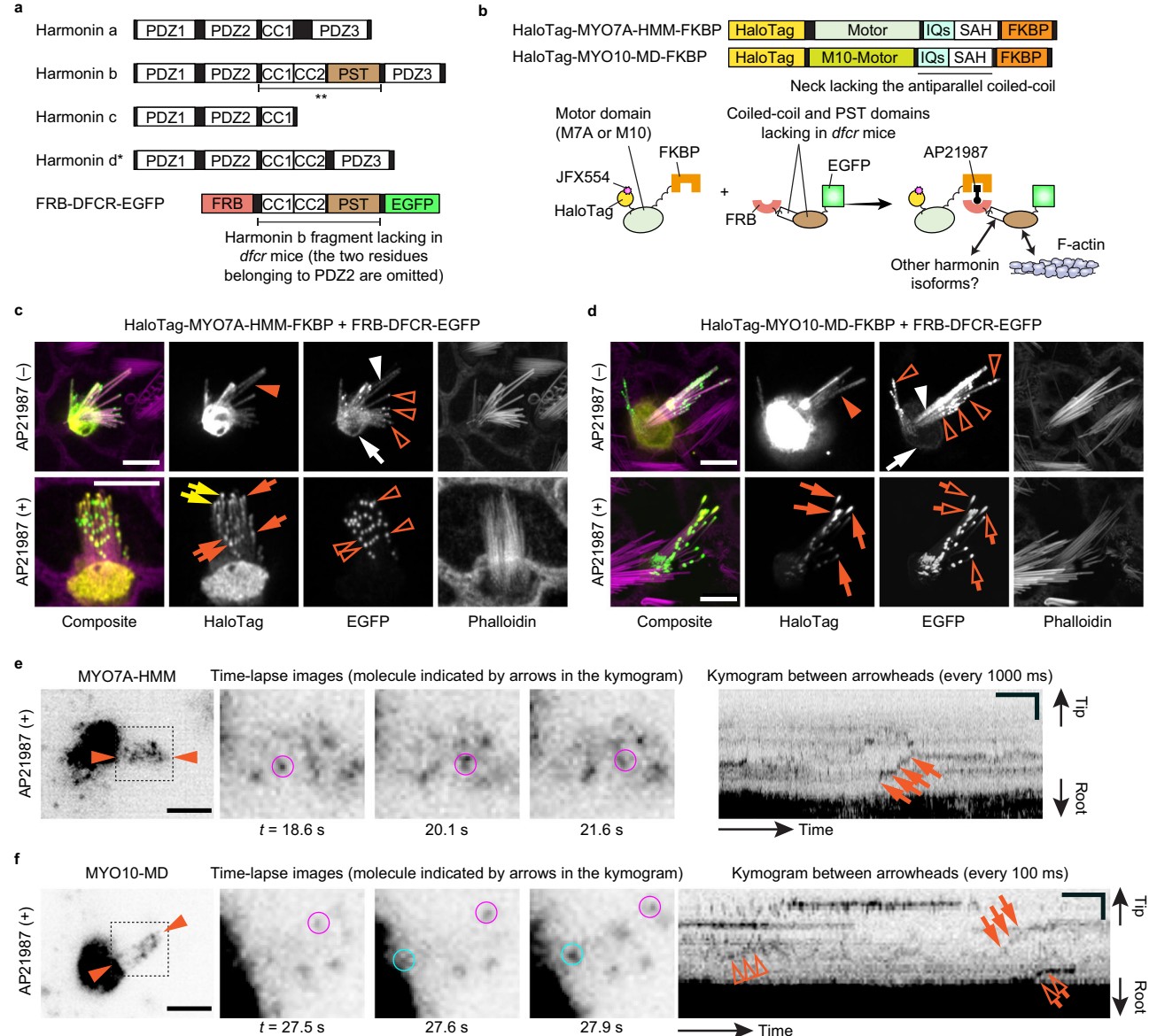

**Fig. 5 | Movement of MYO7A-HMM and MYO10-MD coupled with a harmonin b fragment. a** Major harmonin isoform classes and FRB-DFCR-EGFP, the F-actin anchor used in this study. FRB-DFCR-EGFP contains a harmonin b fragment lacking in *dfcr* mice[123] (double asterisks; residues 296–728 of NM_1163733). Harmonin d was recently identified in the retina[124] (asterisk). **b** Schematic of HaloTag-MYO7A-HMM-FKBP, HaloTag-MYO10-MD-FKBP and their AP21987-dependent coupling with FRB-DFCR-EGFP. The PST domain interacts with F-actin. Two coiled-coil (CC) domains may interact with other harmonin isoforms[80]. **c** Confocal images showing AP21987-dependent coupling between MYO7A-HMM and FRB-DFCR-EGFP. See Fig. 4 for sample preparation. FRB-DFCR-EGFP localizes to stereocilia (white arrowhead) and at the cuticular plate edge (white arrow), with weak accumulation near stereocilia tips (orange open arrowheads). HaloTag-MYO7A-HMM-FKBP distributes diffusely in stereocilia of untreated cells (orange arrowhead) and co-localizes with FRB-DFCR-EGFP in treated cells (orange arrows). A subset of HaloTag-MYO7A-HMM-FKBP localizes to stereocilia tips without dense FRB-DFCR-EGFP puncta (yellow arrows), possibly trafficked there using FRB-DFCR-EGFP as a scaffold. Bars, 5 µm. **d** Confocal images showing relocalization of MYO10-MD via AP21987-dependent

coupling with FRB-DFCR-EGFP. In untreated cells, FRB-DFCR-EGFP localizes similarly to (**c**). MYO10-MD distributes diffusely in stereocilia in untreated cells (orange arrowhead) and accumulates at stereocilia tips with FRB-DFCR-EGFP after AP21987 treatment (orange arrows and open arrows). The amounts of MYO10-MD and FRB-DFCR-EGFP at stereocilia tips increase compared to untreated cells, suggesting that MYO10-MD moves using FRB-DFCR-EGFP as a scaffold. Bars, 5 µm. **e** Single-molecule microscopy of MYO7A-HMM coupled with FRB-DFCR-EGFP. The kymogram shows stepwise, directional movement toward stereocilia tips (arrows). The molecule indicated by arrows is also shown in time-lapse images (magenta circles). AP21987, 500 nM. Single-plane time-lapse, every 1 s. Bars, 5 µm (cell image); 20 s and 2 µm (kymogram). **f** Single-molecule microscopy of MYO10-MD coupled with FRB-DFCR-EGFP. The kymogram shows directional movement toward stereocilia tips (arrows, open arrows and open arrowheads). The molecules indicated by arrows and open arrows in the kymogram are also shown in time-lapse images (magenta and cyan circles, respectively). AP21987, 500 nM. Single-plane time-lapse, every 100 ms. Bars, 5 µm (cell image); 2 s and 2 µm (kymogram).

motor activity is likely inactivated in hair cells until it is unleashed from a backfolded autoinhibitory conformation[53,54] and then dimerized (or perhaps oligomerized) (Fig. 6c). Previous studies report that human MYO7A and *Drosophila* crinkled (myosin 7a) show cargo-mediated dimerization bridged by MYRIP[44] and M7BP[85], respectively. However, RNAseq of FACS-sorted mouse sensory epithelium cells shows that

only a small number of hair cells express *Myrip* (https://shield.hms.harvard.edu/viewgene.html?gene=Myrip). We speculate that components of the tip-link complex, specifically SANS, may be the key to activating MYO7A in stereocilia because processive movement is still detected for MYO7A-ΔSH3-ΔM/F2, a construct that includes SANS binding site (Fig. 3e) but not for MYO7A-HMM-FKBP monomers

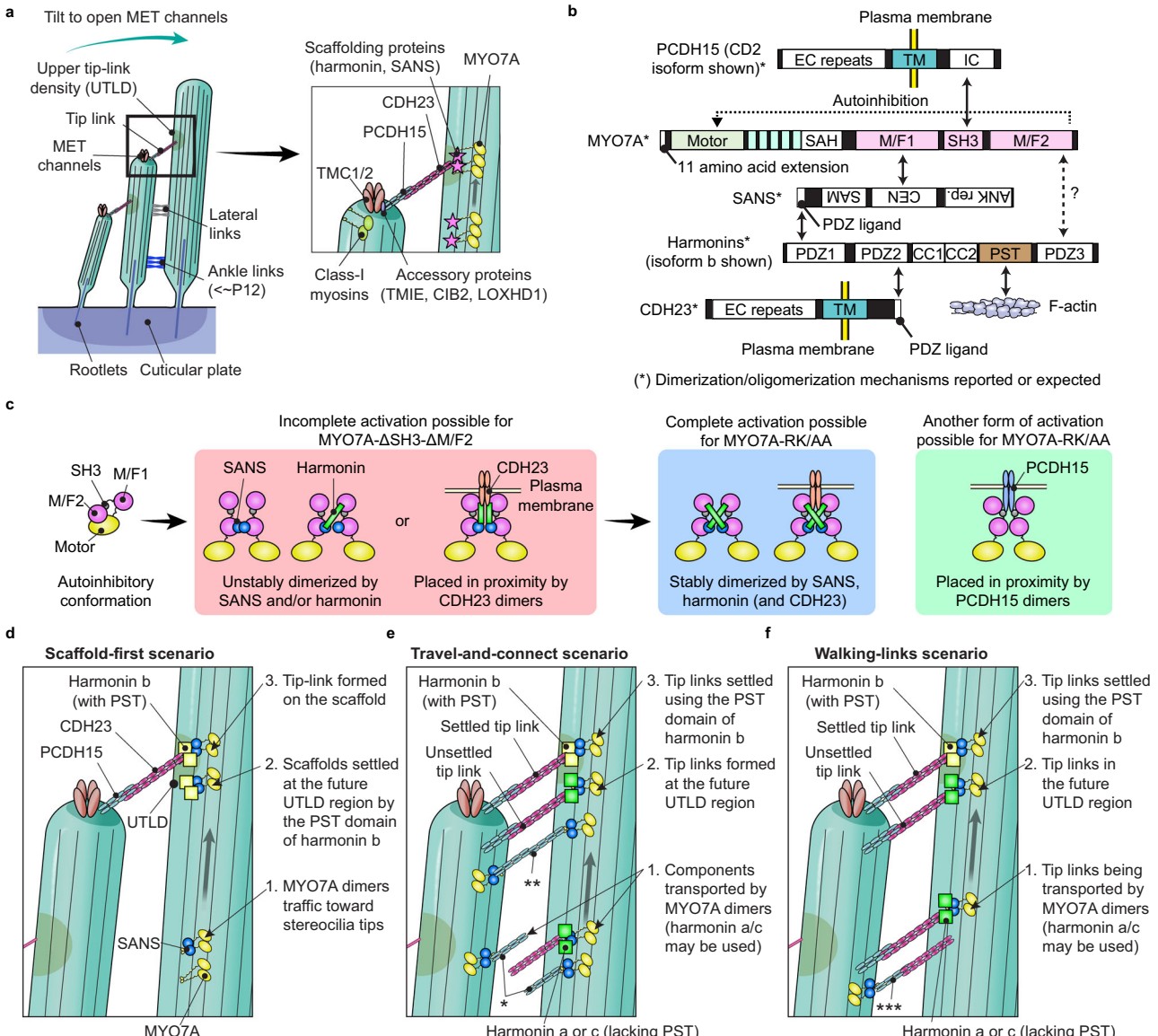

**Fig. 6 | Possible scenarios of MYO7A-driven cargo transport in stereocilia.**
**a** Schematic of stereocilia and MET machinery. MET channels are localized to the distal ends of stereocilia and physically connected to the upper tip-link density (UTLD) of adjacent longer stereocilia. **b** Major interacting partners of MYO7A in stereocilia. SANS and harmonin bridge interactions with other partners. SANS binds to the first MyTH4-FERM domain of MYO7A in an anti-parallel orientation[125]. The PST domain of harmonin b can bind to F-actin[26]. The single α-helix (SAH) of MYO7A has weak dimerization activity[44,85]. SANS, PCDH15 and CDH23 can dimerize with each other[18,19,75] and may keep multiple MYO7A molecules in proximity. Oligo-merization has been reported for harmonin[74]. PCDH15 and CDH23 have one transmembrane motif (TM)[18,19]. The SH3 domain of MYO7A can interact with the intracellular region (IC) of PCDH15 (CD2 isoform shown)[28]. **c** MYO7A activation in stereocilia predicted from the behaviors of MYO7A-RK/AA and MYO7A-ΔSH3-ΔM/F2. From its backfolded autoinhibitory conformation[70], MYO7A could be dimerized by SANS and harmonin as proposed by a previous X-ray crystallography study[86]

(blue panel). MYO7A may also move processively when a PCDH15 dimer brings two MYO7A molecules into proximity (green panel). Considering the low-frequency processive movement of MYO7A-ΔSH3-ΔM/F2, there may be transitional forms (red panel). **d–f** Possible MYO7A-driven tip-link assembly. The scaffold-first scenario (**d**) assumes that MYO7A (with SANS) settles at the future UTLD region via the PST domain of harmonin b and recruits tip-link components. However, MYO7A and SANS lack mechanisms to recognize the future UTLD location. The travel-and-connect scenario (**e**) assumes that either CDH23 or PCDH15 travels toward ste-reocilia tips. The other side may simply diffuse. Harmonin isoforms may switch from harmonin a or c (lacking the PST domain) to harmonin b (with the PST domain), given that MYO7A coupled with harmonin b fragments does not traffic efficiently (see Fig. 5). PCDH15 may form temporary links, as indicated by BAPTA-mediated remodeling experiments[87]. The walking-links scenario (**f**) supported by a previous cryo-electron microscopy study assumes that pre-assembled tip links are transported toward stereocilia tips[88].

(Fig. 2g). This speculation is consistent with a previous study showing that harmonin b does not localize at stereocilia tips in mice with defective MYO7A (*Myo7a^{4626SB/4626SB}*) or SANS (*Ush1g^{js/js}*)[27]. Dimerization of MYO7A may be enhanced by other factors, such as the MYO7A single α-helix (SAH), SANS, PCDH15, CDH23 or harmonin, with their ability to dimerize or oligomerize[18,19,44,74,75,85]. Particularly, harmonin may contribute to the stable dimerization of MYO7A via the interaction between its third PDZ (PDZ3) domain and the MYO7A M/F2 domain[73]

because MYO7A-RK/AA shows processive movement more frequently than MYO7A-ΔSH3-ΔM/F2 (Supplementary Fig. 3c). A previous X-ray crystallography study proposed several forms of MYO7A (or MYO7B) dimers involving SANS (or ANKS4B) and harmonin[86].

Using the current data, we discuss how tip-link components localize in stereocilia. We consider one unlikely scaffold-first scenario (Fig. 6d) and two more likely travel-and-connect and walking-links scenarios (Fig. 6e, f). The scaffold-first scenario expects that the

"scaffold" of MYO7A, SANS and harmonin settles at the future UTLD region and then recruits tip-link components. However, it seems unlikely for MYO7A, SANS and harmonin to know where the future UTLDs would be formed in immature stereocilia. The travel-and-connect and walking-links scenarios expect that CDH23 and PCDH15 reach the future UTLD region, flapping their extracellular portion in the endolymph or with their partners as pre-assembled links, respectively. Remodeling experiments using BAPTA show that new tip links are formed at stereocilia tips after being disrupted by extracellular calcium chelation[87]. This result suggests that tip-link components, CDH23 and PCDH15, are transported although either CDH23 or PCDH15 may simply diffuse to couple with the transported partners. In contrast, a cryoelectron microscopy study using anti-PCDH15 antibodies detects multiple lateral links at the shaft region of stereocilia[88]. These lateral links consist of PCDH15 (~50 nm long) on one side and a longer partner on the other side (~120 nm long), putatively CDH23[88]. Although immunohistochemistry shows MYO7A localizing on the CDH23 side with SANS and harmonin[17–20], the PCDH15 side can be actively transported because (1) many uncoupled PCDH15 molecules are reported near stereocilia tips by a cryoelectron microscopy study[88] and (2) temporary PCDH15-PCDH15 links appear after BAPTA treatment and subsequently change into mature PCDH15-CDH23 links[87] (Fig. 6e, double asterisks). PCDH15 may be transported by MYO3A[76] as well as MYO7A[28] (Fig. 6e, asterisk, illustrated for MYO7A) perhaps with CDH23 on the other side (Fig. 6f, triple asterisks). Stepwise movement of MYO7A coupled with a harmonin b fragment (Fig. 5) can be advantageous for adjusting the position and tension of tip links in mature stereocilia but not for transporting tip-link components over long distances. Thus, harmonin a and c lacking the PST domain may be used in developing stereocilia to transport the CDH23 and UTLD components effectively (Fig. 6e, f).

Future directions of our methodology include analyses of how various unconventional myosins traffic in stereocilia, filopodia and microvilli. For example, it is still uncertain how MYO3A and MYO3B move in F-actin protrusions including stereocilia. The tail of MYO3A lacks a coiled-coil domain for dimerization[47] but has a tail homology domain II (THDII) to directly interact with F-actin and a tail homology domain I (THDI) to interact with F-actin binding protein, ESPN isoform 1 (and ESPNL)[32,33,89]. MYO3B tail has a similar domain structure but lacks THDII[90]. Although inchworm-like movement is hypothesized for these myosins[33,48], single-molecule functional analyses will be necessary, for example, to measure how long and how synchronously their heads and tails crawl with each power stroke. Stepwise movement of MYO7A coupled with a harmonin b fragment (Fig. 5e) suggests that MYO3A and MYO3B may move forward when THDII and/or ESPN isoform 1 dissociate from F-actin. MYO15A has a motor domain with a high duty ratio (~0.5), which is suitable for processive movement as a dimer or an oligomer on F-actin[91] and can be dimerized by CETN2 at the third IQ motif[49]. However, the ability of MYO10-MD to use the plasma membrane and F-actin as a scaffold for directional movement (Figs. 4 and 5) suggests that MYO15A might traffic in stereocilia also using membrane proteins such as ARGRV1 and usherin[92] or F-actin binding proteins such as EPS8 and EPS8L2[29,30,93]. In addition, our data allow us to envision why MYO7A and MYO7B are utilized in stereocilia and microvilli while MYO10 is utilized in filopodia. Exogenous MYO10-MD moves aggressively in stereocilia even as a monomer (Fig. 4), accumulating at stereocilia tips when scaffolded by the plasma membrane or F-actin (Figs. 4 and 5). MYO10-MD induces filopodia-like F-actin protrusions when anchored to the plasma membrane in cultured cells[52]. In our study, MYO7A-HMM does not show processive movement in stereocilia even when tethered to the plasma membrane or F-actin. Coupling with a harmonin b fragment may facilitate the stepwise movement of MYO7A-HMM toward stereocilia tips, but without forming large protein blobs at the tips. Hair cells and intestinal epithelial cells may use class-VII myosins to protect the architecture of stereocilia and microvilli, which are more stable and long-lived than filopodia[94], without inducing unwanted F-actin protrusions.

Single-molecule microscopy is a powerful technique to enable real-time analyses of protein-protein interactions. Recent advances in light-sheet microscopy allow us to apply this technique to stereocilia of live hair cells. Microscopy in the context of live hair cells is essential for hearing research because inner ear hair cells have structures that are never or rarely formed in other organs, such as stereocilia, tip links and ribbon synapses[95], and proteins expressed in limited tissues such as MYO15A and OTOF[96]. In addition to functional analyses of wild-type proteins, our methodology will allow us to detect aberrant behaviors of pathogenic genetic variants, thereby increasing the accuracy of genetic diagnoses based on in silico predictions (e.g., AlphaFold[97] and AlphaMissense[98]). Small molecules may help myosin folding[99], skip premature stop codons[100–102] or suppress dominant-negative effects. In addition to inner ear sensory epithelia, our methodology could be applied to other three-dimensional specimens, including microvilli, primary cilia, kinocilia, migrating cells and neuronal cell layers. Live-cell single-molecule microscopy will continue to elucidate the pathophysiology of diseases such as ciliopathies[103,104], neurodegenerative diseases[105], inflammatory bowel diseases[106,107] and cancer[108], as well as sensorineural hearing loss.

## Methods
### Plasmids and cDNAs
To express HaloTag-fused proteins, the EGFP sequence of the pEGFP-C1 vector (Clontech) was replaced with a HaloTag sequence amplified by polymerase chain reaction (PCR) from the pHTC HaloTag® CMV-neo (Promega). Two silent mutations were introduced to the original HaloTag sequence to disable the XhoI and SalI endonuclease restriction sites. The resulting pHalo2X-C1 vector was used to express HaloTag-fused proteins, including human β-actin derived from pEGFP-actin (Clontech) and myosins. Plasmids encoding HaloTag-fused mouse MYO7A fragments were constructed using the pHalo2X-C1 vector and DNA inserts amplified from a plasmid encoding EGFP-fused mouse *Myo7a* isoform 1 (NM_001256083.1)[29], a gift from Erich Boger, NIDCD. Plasmids encoding HaloTag-fused MYO10 fragments were constructed similarly using pcDNA3.1 Zeo+ EGFP-Myo10noCC-FRB-myc[52], a gift from Matthew Tyska, Vanderbilt University. To express HaloTag-fused myosin fragments with C-terminal FKBP (e.g., HaloTag-MYO7A-HMM-FKBP), a DNA fragment encoding FKBP (LC087168) was amplified from GFP-M7HMM-FKBP[44], a gift from Mitsuo Ikebe at the University of Texas Tyler Health Center, with the C-terminal HA tag. Expression of HaloTag-fused myosin fragments with a C-terminal FRB (e.g., HaloTag-MYO7A-HMM-FRB) was achieved by replacing the FKBP portion of HaloTag-fused myosin fragments with the C-terminal FKBP (e.g., HaloTag-MYO7A-HMM-FKBP) with the FRB sequence of pEGFP-FRB[109] (Addgene #25919). Thus, the C-terminal FKBP and FRB attached to HaloTag-fused myosin fragments have an additional HA tag in this study. A plasmid to express IL2Rα-EGFP-FKBP was constructed from pEGFP-C1 inserting a DNA fragment encoding the Tac antigen (human IL2 receptor alpha subunit; NM_000417.3) amplified from TAC-GFP[110] (Addgene #162494) and the FKBP fragment amplified from GFP-M7HMM-FKBP. A mouse harmonin b fragment (residues 296–728 of NM_01163733) with FRB at the N-terminus and EGFP at the C-terminus (i.e., FRB-DFCR-EGFP) was expressed using a plasmid constructed from pEGFP-N3 (Clontech), an FRB fragment amplified from pEGFP-FRB[109] and a harmonin b fragment amplified from the cDNA of USH1C isoform b4, a gift from Nicolas Grillet at Stanford University. Plasmids pHA-C1 and pHA-N3 to express HA-tagged full-length MYO7A were constructed from pEGFP-C1 and pEGFP-N3 by replacing the EGFP coding sequence with oligonucleotide linkers encoding YPYDVPDYA. The first methionine was added to the HA-tag sequence of pHA-C1 to encode MYPYDVPDYA.

## Animals

All animal experiments were performed in accordance with the National Institutes of Health Guidelines for the Care and Use of Laboratory Animals and approved by the Animal Care and Use Committees at the NIH (No. 1263 to TBF) and SIU (No. 23-027 to TM). Mouse neonates were obtained from pregnant C57BL/6J females purchased from the Jackson Laboratory or obtained from our in-house C57BL/6J colony. Vestibular sensory epithelia were harvested from neonates at postnatal day (P) 2–5 after euthanizing them by decapitation. Mouse sex, age, and tissue origin (utricle or saccule) were documented for potential subgroup analysis. Mice were housed in a standard specific pathogen-free facility under a 12 h light/dark cycle.

## Explant culture and transfection of vestibular sensory epithelia

Inner ear sensory epithelia were cultured and transfected using a Helios® Gene Gun System (Bio-Rad) as previously described with slight modification[56]. Briefly, utricles and saccules of P2–5 mice were isolated in Leibovitz's L-15 medium (Thermo Fisher) after removing the otoliths using a 30 G needle. Isolated sensory epithelia were placed on a glass-bottom dish (MatTek Corporation) coated with rat tail collagen I (A1048301, Thermo Fisher) matrix and cultured in Dulbecco's Modified Eagle Medium/Nutrient Mixture F-12 (DMEM/F12, Thermo Fisher) supplemented with 7% FCS (Atlanta Biologicals) and 20 μg/mL ampicillin (Sigma) at 37 °C in 5% $CO_2$. After culturing for 6–20 h, vestibular hair cells were transfected using a gene gun propelling 1.0 μm gold microcarriers (1652263, Bio-Rad) by helium pulses at 110–115 psi. Gold microcarriers were coated with a 1:1–3:1 mixture of a plasmid encoding a HaloTag-fused protein and pEGFP-C1 (or plasmids encoding EGFP-fused proteins).

## Single-molecule microscopy of live hair cell stereocilia

Vestibular sensory epithelia co-expressing a HaloTag-fused protein and EGFP (or EGFP-fused protein) were labeled using 0.01–0.6 nM JFX554-conjugated HaloTag ligands[57] (gift from Luke Lavis, Janelia Research Campus) diluted in culture medium (DMEM/F12 supplemented with 7% FCS and 20 μg/mL Ampicillin) for 30 min at 37 °C in 5% $CO_2$. Unreacted HaloTag ligands were removed by washing the tissue in the culture medium for 3–5 s three times and by incubating samples at 37 °C in 5% $CO_2$ for up to 6 h until live imaging was performed. Samples on a collagen matrix and the underneath coverslips were detached from glass-bottom dishes, mounted in a 10-cm plastic culture dish on a 1–2 mm droplet of vacuum grease (Fisher Scientific) and then incubated in Leibovitz's L-15 Medium without phenol red (21083027, Thermo Fisher) warmed at 37 °C. For conditional dimerization, the medium was gently removed using plastic transfer pipettes or vacuum suction and replaced with a fresh L-15 medium containing 200 nM AP20187 (Sigma) or 500 nM AP21987 (Takara Bio).

Images were acquired using a custom-made symmetrical dual-view inverted selective plane illumination microscope (diSPIM)[50] installed in a 37 °C incubator (Applied Scientific Instrumentation). The diSPIM installed at National Institute of Biomedical Imaging and Bioengineering was equipped with 40× Nikon CFI APO NIR objectives (0.80 NA, 3.5 mm WD; Nikon), ORCA-Fusion Digital CMOS cameras (C14440-20UP; Hamamatsu), an OBIS 488 nm LX 150 mW Laser (Coherent), an OBIS 561 nm LS 150 mW Laser (Coherent) and a W-VIEW GEMINI Image splitting optic (Hamamatsu) with a 561 nm laser BrightLine single-edge super-resolution/TIRF dichroic mirror (Semrock), a 525/50 nm BrightLine single-band bandpass filter (Semrock) and a 568 nm EdgeBasic best-value long-pass edge filter (Semrock). The diSPIM available at Southern Illinois University omits the W-VIEW GEMINI Image splitting optic and has a ZT488/561rpc dichroic mirror (Chroma) and a ZET488/561m bandpass filter (Chroma). Microscopes were controlled by the Micro-Manager (https://micro-manager.org/) plugin for ImageJ[111–113].

HaloTag-fused myosin molecules were imaged by single-plane time-lapse acquisition every 0.1 to 1 s, illuminating the 561-nm laser at approximately 0.2 kW/cm² for 100 ms per frame. Sample drift was corrected using custom Python scripts available at the first author's GitHub repository (http://github.com/takushim/momomagick), which implement phase-only correlation with subpixel matching[114] and least-squares image matching[115]. Fluorescent puncta were manually tracked using our custom Python script with a graphical user interface (http://github.com/takushim/momotrack). Kymograms were generated using the Multi Kymograph function on the Fiji platform[116]. The entire architecture of stereocilia was visualized using fluorescence from co-expressed EGFP or EGFP-fused proteins by a volume scan of 0.5-μm thickness illuminating the 488 nm laser at approximately 0.01 kW/cm² for 100 ms per slice.

Non-fused HaloTag and HaloTag-actin molecules were imaged by a single volume scan of 0.5-μm thickness for Fig. 1b–d, and by single-plane time-lapse acquisition for Fig. 1e, f, and Supplementary Fig. 1 using the excitation conditions for myosin molecules. Intensity profiles of fluorescent puncta were obtained using the profile_line function of scikit-image (https://scikit-image.org/) and quadratically interpolated using the interp1d function of SciPy (https://scipy.org/). These line profiles were normalized between 0 and 1 and averaged after being aligned to the intensity peaks found by the find_peaks function (SciPy). The point spread function of the 40× objective lens was calculated using the PSF generator (https://bigwww.epfl.ch/algorithms/psfgenerator/) based on the Born & Wolf 3D Optical Model[117,118] where Refractive index immersion = 1.33, Wavelength = 576 nm[57] and NA = 0.8. Photobleaching was measured using the cuticular plate of hair cells expressing HaloTag-actin that were fixed in 4% paraformaldehyde (PFA; Electron Microscopy Sciences) in PBS for 30 min at room temperature (RT, 20–25 °C), washed three times in PBS for 5 min and immersed in PBS warmed at 37 °C.

## Fluorescence histochemistry

Explant cultures of vestibular sensory epithelia were fixed in 4% PFA in PBS for 30 min at RT and washed in PBS. For conditional dimerization, samples were treated with 100 nM AP20187 (Sigma) or 500 nM AP21987 (Takara Bio) for 2 h before fixation. The concentration of AP20187 was lowered because 200 nM AP21087 caused a strong accumulation of MYO7A-HMM dimers at stereocilia tips and often severely damaged the architecture of stereocilia. Fixed samples were permeabilized and blocked in PBS containing 1% bovine serum albumin (BSA) and 0.2% Triton X-100 (PBS-BSA-Tx) for 20–30 min at RT. HaloTag-fused proteins were visualized using 200 nM JFX554-conjugated HaloTag-ligands[57] (gift from Luke Lavis) in PBS with 0.2% Triton X-100 (PBS-Tx) for 1 h at RT. F-actin was visualized by supplementing the HaloTag ligand solution with 10–30 nM Alexa Fluor™ Plus 405 Phalloidin (Thermo Fisher). For immunohistochemistry of the HA tag, samples were blocked and permeabilized in PBS-BSA-Tx for 1 h at RT and then incubated with HA-Tag (C29F4) Rabbit mAb #3724 (Cell Signaling Technology) diluted 1:500 in PBS-Tx for 2 h at RT. After washing in PBS for 5 min three times, the anti-HA tag antibody was visualized using an Alexa Fluor 568-conjugated goat anti-rabbit IgG (H + L) antibody (A-11036, Thermo Fisher Scientific) diluted 1:500 in PBS-Tx for 1 h at RT. F-actin was counterstained with 10–30 nM Alexa Fluor™ Plus 405 Phalloidin. The anti-HA tag antibody was validated using HeLa cells ectopically expressing HA-tagged β-actin. Confocal images were acquired using a Zeiss LSM880 Airyscan confocal microscope equipped with a 63×/1.4 N.A. objective.

## Statistics and reproducibility

Fluorescent puncta in Fig. 1c were classified using the Gaussian Mixture model[60] available in the scikit-learn package (https://scikit-learn.org/). The rate constant (K) and half-life ($T_{1/2}$) in Supplementary Fig. 1 were determined by fitting a one-phase decay model using GraphPad Prism.

These values were compared against a null hypothesis that one model explains all data using the extra-sum-of-squares F-test implemented in GraphPad Prism[119] followed by the correction of $P$ values using the conventional Bonferroni method[120]. The frequency, velocity and run lengths of directionally moving molecules were compared using a two-sided Student's $t$-test or one-way analysis of variance (ANOVA) followed by the post-hoc Tukey test implemented in GraphPad Prism. A directionally moving molecule was defined as a molecule showing a trajectory in the same direction for more than three frames in a kymogram. Molecules moving back and forth in a stereocilium are presumed to be in random molecular motion and were excluded from tracking. The frequency of directionally moving molecules was calculated using time-lapse images acquired for more than 500 frames. In Supplementary Fig. 3a, the ratio of hair cells accumulating HaloTag-fused MYO7A at stereocilia tips was compared using the Pairwise comparisons using Fisher's exact test implemented in the fisher.multicomp function in R.

All single-molecule microscopy and histochemistry experiments were performed using at least three different cells from three different sensory epithelia (Fig. 1b, e, f; Fig. 2c, d, g; Fig. 3b–e; Fig. 4b–e; Fig. 5c–f).

### Reporting summary

Further information on research design is available in the Nature Portfolio Reporting Summary linked to this article.

## Data availability

The numerical data underlying all graphs in this study are provided in the Source Data file. Representative images from single-molecule imaging experiments are shown in the main figures and are also available as Supplementary Movies. Raw single-molecule and confocal imaging data have been deposited in Zenodo (https://doi.org/10.5281/zenodo.16233093). The custom plasmids used in this study will be distributed via Addgene. Image acquisition and analysis procedures are described in the Methods section. All image data used in this manuscript are available from the first author upon request and under a data use agreement. Source data are provided with this paper.

## Code availability

Custom Python scripts used in this study have been archived on Zenodo and assigned the following DOIs: momomagick[121] (https://doi.org/10.5281/zenodo.16503793) and momotrack[122] (https://doi.org/10.5281/zenodo.16503757). For continued development and the latest updates, the code is also available at the first author's GitHub repository (https://github.com/takushim).

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

## Acknowledgements

We thank Drs. Dennis Winkler and Mhamed Grati for valuable comments; Elizabeth Bernhard, Sherly Michel and Alexander Callahan for managing mouse colonies at the National Institute on Deafness and Other Communication Disorders; Erich Boger for providing plasmids encoding myosins and cargo proteins; Phil Jensik for mentoring the first author during the launch of his laboratory at Southern Illinois University (SIU); Buffy Ellsworth and Pratyusa Das for sharing their animal housing space during the startup phase of the first author's laboratory at SIU; Jiji Chen and Min Guo for comments on single-molecule microscopy and image processing; Ichiro Fujii for comments on statistical analyses; Mikaela Aholt for analyzing a subset of the image data during her laboratory rotation; Steve Saltekoff for assisting with the setup a diSPIM microscope at SIU; and Erina He for her beautiful diagrams. The diSPIM images were acquired in the Advanced Imaging and Microscopy Resource of the National Institute of Biomedical Imaging and Bioengineering and in the first author's laboratory at SIU. This article is subject to Howard Hughes Medical Institute (HHMI)'s Open Access to Publications policy. HHMI laboratory heads have previously granted a non-exclusive CC BY 4.0 license to the public and a sub-licensable license to HHMI for their research articles. Pursuant to those licenses, the author-accepted manuscript of this article can be made freely available under a CC BY 4.0 license immediately upon publication. T.M., I.A.B., Y.I., S.M.A., and T.B.F. were supported (in part) by NIDCD intramural research funds DC000039 (to T.B.F.). H.D.V. and H.S. were supported by NIBIB intramural research funds (to H.S.). H.S. was also supported by HHMI. T.M. was supported by JSPS Overseas Research Fellowships. T.M., J.M., and M.S. were supported by start-up funds from SIU School of Medicine (to T.M.), funding from the Office of the Vice Chancellor for Research at SIU (to T.M.) and the R00 Pathway to Independence Award (1R00DC019949 to T.M.).

## Author contributions

T.M.: Conceptualization, Methodology, Software, Validation, Formal analysis, Investigation, Writing – Original draft, Visualization, Project administration and Funding acquisition. H.D.V.: Methodology and Investigation. I.A.B.: Methodology, Writing – Review & Editing and Visualization. J.M.: Investigation and Visualization. M.S.: Investigation, Validation, Writing – Review & Editing, Formal analysis and Visualization. Y.I.: Validation and Writing – Review & Editing. S.A.: Validation and Writing – Review & Editing. N.H.: Conceptualization and Methodology. H.S.: Conceptualization, Methodology, Writing – Review & Editing, Supervision and Funding acquisition. T.B.F: Writing – Review & Editing, Supervision, Project administration and Funding acquisition.

## Competing interests

The authors declare no competing interests.
