## [Transparent Peer Review file · Nature Communications]

Single-molecule fluorescence microscopy reveals regulatory mechanisms of MYO7A-driven cargo transport in stereocilia of live inner ear hair cells

Corresponding Author: Dr Takushi Miyoshi

Version 0:

Reviewer comments:

Reviewer #1

(Remarks to the Author)

In this manuscript, Miyoshi et al seek to image the trafficking of fluorescently labelled myosins expressed in inner ear hair cell explants. In particular, the authors are interested in the motor protein Myo7A, which is mutated in a number of human sensory disorders including DFNA11/DFNB2 deafness and USH1B deaf-blindness. The group first develops a protocol to overcome the technical limitations that are encountered in trying to image molecules found in membrane protrusions on the surface of epithelial cells. Using a dual-view light-sheet microscope, the authors are able to successfully resolve single fluorescently-labelled molecules in stereocilia. The group then utilizes an inducible dimerization system to observe that an artificially-dimerized truncated fragment of Myo7A can exhibit processive movement towards the (+) ends of stereocilia. In contrast, a full-length Myo7A construct did not exhibit trafficking on its own, suggesting that it may need to be activated before exhibiting motility (perhaps by cargo). Mutating or removing autoinhibitory sequences from the Myo7A tail did allow for limited instances of processive motility to be observed. Artificially coupling a truncated fragment of Myo7A to plasma membrane using a heterodimerization system allowed for motility, though motility was different in nature compared to the homodimerized Myo7A fragment. In a similar manner, artificially tethering Myo7A to the F-actin cytoskeleton using the PST domain of harmonin-b (a native cargo molecule of Myo7A) also allowed for motility, though the trajectories of these myosins in kymographs were different from those of Myo7A homodimer fragments.

This is a well-written manuscript from the Miyoshi/Friedman labs that will be of interest to a wide audience given the relevance to not only those interested in sensory perception, but also myosins and imaging modalities in general. The studies are presented in a manner that is easy to follow and include all the appropriate controls and analysis. I have a few questions/comments that I would like the authors to address.

Major Comments:

1) Pg 8. Line 240. Studies using the RK/AA and MYO7A- Δ SH3- Δ M/F2 state that these MYO7A mutants accumulated at stereocilia tips in "a few transfected hair cells". Can the authors provide some numbers for this? How does this compare to the forced homodimer Myo7A construct?

2) The localization of the FRB-PST-EGFP construct is a little surprising. When expressed in a heterologous cell system (eg HeLa cells), this harmonin-b fragment appears to be quite potent at binding and cross-linking F-actin (huge bundles of F-actin typically appear in the cytoplasm of transfected cells- this has been seen in many studies). In contrast, very little of the FRB-PST-EGFP on its own appears to bind up in the stereocilia. Also, it appears to bind mostly at the stereocilia tips. It really doesn't look like the PST domain "anchors" to the F-actin core of stereocilia very well. Can the authors comment on this?

To my knowledge, I don't think its known whether harmonin-b bundles actin in a parallel or antiparallel fashion. If it is an antiparallel-bundler and prefers to associate with antiparallel-bundled actin, it might not be functioning as an "actin tether" to the stereocilia core as the authors envision. It is also worth noting, that in order for harmonin-b to bundle actin, it needs to either have two or more inherent F-actin binding sites, or needs to dimerize (or oligomerize). If the PST domain is functioning as an oligomer, this also complicates the interpretation of the authors data. The authors should comment on this.

3) Pg. 10 line 318-319. It is unclear to me what the authors mean by "These results suggest that myosin molecules tethered to F-actin move only when the tail is released from F-actin." If the PST domain doesn't dimerize the myosin, and it lets go of the F-actin track, how is the monomeric myosin able to exhibit processive motility? Can the authors elaborate on what their idea is?

4) The manuscript is well-written, though somewhat long in places (intro and discussion). I think both these sections could be shortened without losing any valuable information that needs to be communicated.

Minor comments:

1) The authors report they are using a mouse Myo7A splice isoform (transcript NM_001256083.1). However, the cartoon diagram in Fig. 2A appears to contain the 11 amino acid N-terminal extension found in the canonical splice isoform of Myo7A. Looking at the NM_001256083.1 transcript sequence, this splice isoform does not have this N-terminal extension. Can the authors double check the isoform that they used for this study?

The NM_001256083.1 splice isoform also deviates from the canonical Myo7A isoform in that it is missing ~38 amino acids at the end of FERM1. To my knowledge, it is not known whether this would influence cargo binding (eg Sans).

Reviewer #2

(Remarks to the Author)

Reviewer #3

(Remarks to the Author)

General comments:

The authors address an important question in the field: unconventional myosins are poorly understood, particularly in a live-cell context, and the authors correctly point out that traditional single-molecule approaches are not suitable for stereocilia as movement occurs parallel to the plane of imaging. To address this problem, the authors exploit a diSPIM approach, which was established in their previous paper (Miyoshi et al, Cell Rep 2021), to image tagged myosin in live stereocilia. The approach itself is therefore not novel but its use in imaging myosin at the single-molecule level is.

Methods are well reported although the number of biological and technical replicates need to be more clearly stated and there needs to be more analysis of the data (see later).

However, the work itself does not meet the expected standards in the field for 3 primary reasons:

1) Data quality is not always clear

2) There is little to no evidence of repeats, with $n = 1$ for the majority of the claims.

3) There are no statistics and therefore no evidence to support any of the conclusions made

These are major issues that prohibit publication.

Specific comments:

Given the lack of repeats or statistics, it is hard to write specific comments. However, here are some (hopefully) constructive feedback for the authors to turn this work into a publication with MAJOR points indicated (the rest of the points are MINOR).

Introduction

- Paragraph 2 on the single-molecule microscopy breaks the flow of the intro. I suggest placing it after introducing the biological system i.e. after introducing myosin and before the final paragraph of the intro and use that paragraph to explain why single-molecule imaging is the best way to understand myosin.

- For someone who is not an expert on MET channels, please explain possible ways in which myosin could affect the MET channels before you then show data highlighting that one model is correct - what are the possible ways they could act before you collect your data e.g. they could tether the tip link complex, transport channels to the tip etc.

Result 1: Single-molecule microscopy in stereocilia of live hair cells

- Fig. 1a Plasmids generated are a lot of work and great to see that it all works. Approach to get plasmids into these cells using the Helios® gene-gun also seemed really successful and cool. Please provide the efficiency of the approach (% of cells labelled) so others can assess using it.

- (MAJOR) Fig. 1b In the movies, there are some parts of cells with a lot of over-labelling and it can be very hard to assess if you are truly looking at single molecules or clumps of molecules. This makes it hard to use automated software to pick out the dots and the hand picking of dots is really not ideal. Why not use photoactivatable PA-JF549 to reduce overlap of molecules so you can more sure that you are looking at single molecules and so you can image more molecules per cell without worrying about labelling density? Is it because there is no 405 nm photoactivation laser available?

- (MAJOR) Fig. 1c What is the background level when you don't add dye? You can't say puncta were classified into 2 populations because you split the dots in half yourself. You'll need to use a classifier e.g. gaussian mixture model with BIC analysis to show this.

- Fig. 1d You will need to get the PSF of all your molecules and then make error bars or a 95 % confidence interval from the molecules.
- (MAJOR) Authors show single-step photobleaching traces as evidence molecules disappear in a single frame which is good but the structures also move in the videos so a fixed sample will be needed to show this is not movement of the cilia. This will also give the precision of the data and the photobleaching rate that should be reported in the text and shown as a dotted line when plotting movement parameters.
- (MAJOR) Authors say that, at 1 second time resolution, diffusing molecules are there one frame but actin molecules last ... the video does show this but the conclusion will need $n = 3$ cells at least. It will also need track lengths to be calculated for the various samples (ideally with a fixed cell control) and compared using a stats test, especially as some of the diffusing molecules do get stuck so the relative difference will be important.

Result 2: Visualization of directional movement of MYO7A dimers in stereocilia

- (MAJOR) Authors say MYO7A shows “continuous trajectories consistent with processive movements” that “move toward the barbed ends of unidirectional F-actin bundles in” but these conclusions cannot be made without more trajectories and statistics. Kymograms and videos show many dots that don't move at all (Fig.2d-f) so how are molecules chosen to calculate the movement? Are static molecules not shown or are the non-moving dots considered to be background? Surely percentage of non-moving molecules should also be reported. Statistics are also key to report movement e.g. compare molecules +/- AP20187, measure velocity and show statistical changes and report p-values (especially as molecules are sometimes not moving at all). To report directionality, the standard way is to compare the angle from one displacement to the next (or from every n displacements to the next n displacements if movement is slow) and calculate the angle difference e.g. using Circstat on MATLAB and then use a fold anisotropy metric $f(180/0)=P(180\pm 30^\circ/0\pm 30^\circ)$ to define how many-fold more likely a molecule is to make a step backwards compared to a step forward +/- AP20187. You could also define the angle as towards or away from the tip?

Result 3: Constitutively active MYO7A mutants move directionally in stereocilia

- (MAJOR) Authors say “HaloTag-fused full-length MYO7A did not show directional movements at a detectable frequency in stereocilia”, “Movements of MYO7A-RK/AA were directional and processive resembling the movements of MYO7A-HMM dimers” etc but as above, how many molecules, where are movement and directionality quantification and statistics?

Result 4: Distinct behavior of MYO7A and MYO10 anchored to the plasma membrane

- (MAJOR) Authors again make a series of conclusions without stats e.g. “Movements of membrane-anchored MYO7A-HMM were different from MYO7A-HMM dimers or constitutively active MYO7A mutants”, “Directional movements of MYO10-MD indicate that the restricted movements of MYO7A-HMM are derived from the kinetic differences between the motor domains of MYO7A and MYO10” (are they? What is the p-value?) They do quantify speed this time and show stdev which means difference is significant (although it is best to report the p-value), which is great, but length of track is also an important parameter. It looks like the AP21987 slows it down so then it takes longer to get to the tip i.e. slower and longer tracks which lead to the same total distance covered.

- Fig. 4e shows, as the authors explain, that molecules can move both to the tip and back so quantification of direction will be important here and it would be interesting to compare MYO7A and MYO10 +/- AP21987 using a directional metric to show if it goes from random diffusion to directional motion ... after all, we don't see as many tracks in MYO7A that go away from the tip i.e. it could be more directional.

Result 5: Step-wise movements of MYO7A and MYO10 when tethered to F-actin

- “Movements of MYO7A-HMM molecules tethered to F-actin were step-wise as observed for those anchored to the plasma membrane. MYO10-MD molecules also showed step-wise movements after the AP21987 treatment” Again this needs to be quantified e.g. number of consecutive movements in the same direction that are above the precision limit i.e. that are not static. A histogram of this number of consecutive movements will show whether the steps are always the same size and quantify the distance taken per step. This histogram can then be compared to the previous datasets that are not stepwise to make the point that some traces are more stepwise than others
- “These observations are not consistent with an “inchworm-like” movement proposed by others because the step-sizes were 100–200 nm” – Could there be inchworm-like movement occurring at faster timescales than the time resolution of the videos?

Reviewer #4

(Remarks to the Author)

In the present manuscript, the authors utilized live-cell single-molecule fluorescence microscopy to examine MYO7A transport in the stereocilia of inner ear hair cells. Using various MYO7A truncations and mutants, they showed that MYO7A traffics in the stereocilia as dimers (or oligomers), and its movement is restricted when scaffolded by the plasma membrane or stereociliary F-actin core. In general, it is an interesting and well-performed work, and is of significance to the related field. Similar strategies could be used to explore the transport of other important proteins in the stereocilia. This reviewer has some concerns and suggestions that are listed below.

1. The data in the present manuscript did not answer how native MYO7A is transported to the stereocilia tips. Is native, full-length MYO7A transported when attached to plasma membrane? Coupled to the F-actin core? Or both? The authors showed that “Halo Tag-fused full-length MYO7A did not show directional movements at a detectable frequency in stereocilia of vestibular hair cells”, and explained that it might be because “full-length MYO7A takes a backfolded autoinhibitory conformation between the tail and motor domains” (line 229-232). It's interesting why endogenous MYO7A can release this autoinhibition in some way, and somehow Halo Tag-fused full-length MYO7A cannot. Nevertheless, the authors showed that

MYO7A RK/AA mutant can move directionally in the stereocilia. What is the velocity of RK/AA mutant movement? Could the behavior of MYO7A RK/AA mutant represent that of the native MYO7A?

2. Shin and colleagues reported that there are various MYO7A isoforms with different N-termini (MYO7A-S and MYO7A-C, Li et al, NC 2020). Do these different N-termini affect the transport of MYO7A to the tips of stereocilia?

3. Several deafness-mutations in the motor region of MYO7A have been identified. Does any of these mutations affect the transport of MYO7A to the stereocilia tips?

4. The models presented in Figure 6c-e are nice. Could the authors clearly state which model their data support the most?

Version 1:

Reviewer comments:

Reviewer #1

(Remarks to the Author)

I thank the authors for addressing my comments in their rebuttal. The modifications made to the revised manuscript have helped strengthen their conclusions.

Reviewer #2

(Remarks to the Author)

Reviewer #3

(Remarks to the Author)

We thank the authors for diligently going through and addressing all the reviewer's comments. The paper is now robust and we have no further comments.

Reviewer #4

(Remarks to the Author)

The revised manuscript is greatly improved. The authors addressed most of the questions. However, it's still not clear to this reviewer why Halo Tag-fused full-length MYO7A cannot show directional movements at a detectable frequency in stereocilia. The authors speculate that "some endogenous factors, such as components of the tip-link complex, unleash MYO7A from an autoinhibitory state and mediate dimerization (oligomerization) of MYO7A molecules." How does this apply only to endogenous full-length MYO7A, but not Halo Tag-fused full-length MYO7A? Could the authors elaborate more on this?

In page 14, line 427-428, "Fig. 6d" should be "Fig. 6e"; in line 429, "Fig. 6e" should be "Fig. 6f".

Version 2:

Reviewer comments:

Reviewer #4

(Remarks to the Author)

The authors addressed all my concerns. This reviewer has no more questions.

March 19, 2025

Dear Reviewers,

We are grateful for your valuable comments and provide point-by-point responses below. Fig. 4d, Fig. 5c, Movie S4 and Movie S7 were replaced with new images. All changes to the original manuscript are visible in Track Changes in a **Word** file submitted as a supplement.

Reviewer #1 (Remarks to the Author):

“In this manuscript, Miyoshi et al seek to image the trafficking of fluorescently labelled myosins expressed in inner ear hair cell explants. In particular, the authors are interested in the motor protein Myo7A, which is mutated in a number of human sensory disorders including DFNA11/DFNB2 deafness and USH1B deaf-blindness. The group first develops a protocol to overcome the technical limitations that are encountered in trying to image molecules found in membrane protrusions on the surface of epithelial cells. Using a dual-view light-sheet microscope, the authors are able to successfully resolve single fluorescently-labelled molecules in stereocilia. The group then utilizes an inducible dimerization system to observe that an artificially-dimerized truncated fragment of Myo7A can exhibit processive movement towards the (+) ends of stereocilia. In contrast, a full-length Myo7A construct did not exhibit trafficking on its own, suggesting that it may need to be activated before exhibiting motility (perhaps by cargo). Mutating or removing autoinhibitory sequences from the Myo7A tail did allow for limited instances of processive motility to be observed. Artificially coupling a truncated fragment of Myo7A to plasma membrane using a heterodimerization system allowed for motility, though motility was different in nature compared to the homodimerized Myo7A fragment. In a similar manner, artificially tethering Myo7A to the F-actin cytoskeleton using the PST domain of harmonin-b (a native cargo molecule of Myo7A) also allowed for motility, though the trajectories of these myosins in kymograms were different from those of Myo7A homodimer fragments.

This is a well-written manuscript from the Miyoshi/Friedman labs that will be of interest to a wide audience given the relevance to not only those interested in sensory perception, but also myosins and imaging modalities in general. The studies are presented in a manner that is easy to follow and include all the appropriate controls and analysis. I have a few questions/comments that I would like the authors to address.”

Major Comments:

1) Pg 8. Line 240. Studies using the RK/AA and MYO7A- Δ SH3- Δ M/F2 state that these MYO7A mutants accumulated at stereocilia tips in “a few transfected hair cells”. Can the authors provide some numbers for this? How does this compare to the forced homodimer Myo7A construct?

Response:

Thank you for your comments. We compared the ratio of cells accumulating MYO7A at stereocilia tips between HaloTag-MYO7A-RK/AA, HaloTag-MYO7A- Δ SH3- Δ M/F2 and HaloTag-fused full-length MYO7A. We used full-length MYO7A as a control to show how strongly RK/AA and Δ SH3- Δ M/F2 are activated by the disrupted autoinhibition.

Our statistical analyses show that MYO7A-RK/AA accumulates at stereocilia tips significantly more frequently than full-length MYO7A but MYO7A- Δ SH3- Δ M/F2 does not. We included the numbers of cells in Fig. S3a and modified Fig. 3 to show cells accumulating Δ SH3- Δ M/F2 at stereocilia tips and cells lacking this phenotype. In the

Results section, the first paragraph of the “Processive movements of constitutively active MYO7A mutants in stereocilia” subsection was rephrased as follows:

Page 8, Line 236:

“Confocal microscopy shows that HaloTag-MYO7A-RK/AA often accumulates at stereocilia tips, while full-length MYO7A diffusely distributes in stereocilia (Fig. 3b, representative images shown). HaloTag-MYO7A- Δ SH3- Δ M/F2 distributes diffusely in stereocilia but seldomly accumulates at stereocilia tips (Fig. 3c). Pairwise comparisons using Fisher’s exact test show $P = 0.042$ between full-length MYO7A and MYO7A-RK/AA on the formation of protein blobs at stereocilia tips (Fig. S3a).”

2) The localization of the FRB-PST-EGFP construct is a little surprising. When expressed in a heterologous cell system (eg HeLa cells), this harmonin-b fragment appears to be quite potent at binding and cross-linking F-actin (huge bundles of F-actin typically appear in the cytoplasm of transfected cells- this has been seen in many studies). In contrast, very little of the FRB-PST-EGFP on its own appears to bind up in the stereocilia. Also, it appears to bind mostly at the stereocilia tips. It really doesn’t look like the PST domain “anchors” to the F-actin core of stereocilia very well. Can the authors comment on this?

To my knowledge, I don’t think its known whether harmonin-b bundles actin in a parallel or antiparallel fashion. If it is an antiparallel-bundler and prefers to associate with antiparallel-bundled actin, it might not be functioning as an “actin tether” to the stereocilia core as the authors envision. It is also worth noting, that in order for harmonin-b to bundle actin, it needs to either have two or more inherent F-actin binding sites, or needs to dimerize (or oligomerize). If the PST domain is functioning as an oligomer, this also complicates the interpretation of the authors data. The authors should comment on this.

Response:

Thank you for your comments. We agree that the localization of FRB-DFCR-EGFP (renamed from FRB-PST-EGFP) at stereocilia tips cannot be completely explained by its interaction with F-actin. However, we consider that FRB-DFCR-EGFP’s interaction with F-actin is suggested by its localization at the cell edge (Fig. 5c, white arrow) and by directional movement of MYO10-MD toward stereocilia tips using this fusion protein as a scaffold (Fig. 5f). Confocal microscopy shows dense localization of FRB-DFCR-EGFP near stereocilia tips, which may be explained by its interaction with the coiled-coil domain of endogenous harmonin. In the Results section, we clarified this discussion by rephrasing the first paragraph of the subsection “Possible step-wise movements of MYO7A coupled with a harmonin b fragment” as follows:

Page 11, Line 331:

“Interestingly, FRB-DFCR-EGFP shows accumulation at or near stereocilia tips (Fig. 5, c and d, orange open arrowheads), which may co-localize with endogenous harmonin as antiparallel dimers using the second coiled-coil (CC2) domain in our fragment⁸⁰. After AP21987 treatment, HaloTag-MYO7A-HMM-FKBP co-localizes with FRB-DFCR-EGFP at or near stereocilia tips (Fig. 5c, orange arrows) indicating that AP21987 couples MYO7A-HMM with DFCR fragments. Part of HaloTag-MYO7A-HMM-FKBP may be able to traffic toward stereocilia tips using the DFCR fragment as a scaffold since HaloTag-MYO7A-HMM-FKBP sometimes localizes at stereocilia tips without dense FRB-DFCR-EGFP puncta (Fig. 5c, yellow arrows). HaloTag-MYO10-MD-FKBP forms protein blobs at stereocilia tips with FRB-DFCR-EGFP after AP21987 treatment (Fig. 5d, orange

arrows and open arrows) suggesting that MYO10-MD molecules can move toward stereocilia tips using DFCE fragments as a scaffold presumably to tether the MYO10-MD C-terminus to F-actin.”

3) Pg. 10 line 318-319. It is unclear to me what the authors mean by “These results suggest that myosin molecules tethered to F-actin move only when the tail is released from F-actin.” If the PST domain doesn’t dimerize the myosin, and it lets go of the F-actin track, how is the monomeric myosin able to exhibit processive motility? Can the authors elaborate on what their idea is?

Response:

We apologize for this confusion. Without experimental support, some myosins (e.g., MYO3A and MYO3B) are considered to move on F-actin^{1,2} like an inchworm. However, our single-molecule microscopy study shows that HaloTag-MYO7A-HMM may show step-wise movements when coupled with a harmonin b fragment. This finding suggests that myosins whose tails are tethered to F-actin may move toward the barbed ends only when their tails are released from F-actin. In the revised manuscript, we elaborate on this idea and offer our speculation in the Discussion section as follows:

Page 14, Line 435:

“For example, it is still uncertain how MYO3A and MYO3B move in F-actin protrusions including stereocilia. The tail of MYO3A lacks a coiled-coil domain for dimerization⁴⁷ but has a tail homology domain II (THDII) to directly interact with F-actin and a tail homology domain I (THDI) to interact with F-actin binding protein, ESPN isoform 1 (and ESPNL)^{32,33,89}. MYO3B’s tail has a similar domain structure but lacks THDII⁹⁰. Although “inchworm-like” movements are hypothesized for these myosins^{33,48}, single-molecule functional analyses will be necessary, for example, to measure how long and how synchronously their heads and tails crawl on each power stroke. Step-wise movements of MYO7A coupled with a harmonin b fragment (Fig. 5e) suggest that MYO3A and MYO3B may move forward when THDII and/or ESPN isoform 1 dissociates from F-actin.”

4) The manuscript is well-written, though somewhat long in places (intro and discussion). I think both these sections could be shortened without losing any valuable information that needs to be communicated.

Response:

We agree and have shortened the Introduction and Discussion sections. Several sentences not essential for the main story were removed. The paragraphs in the Introduction section were reorganized according to a detailed comment of Reviewer # 3. Our edits are visible in Track Changes. The revised Introduction and Discussion have approximately 200 and 400 fewer words, respectively.

Minor comments:

1) The authors report they are using a mouse Myo7A splice isoform (transcript NM_001256083.1). However, the cartoon diagram in Fig. 2A appears to contain the 11 amino acid N-terminal extension found in the canonical splice isoform of Myo7A. Looking at the

NM_001256083.1 transcript sequence, this splice isoform does not have this N-terminal extension. Can the authors double check the isoform that they used for this study? The NM_001256083.1 splice isoform also deviates from the canonical Myo7A isoform in that it is missing ~38 amino acids at the end of FERM1. To my knowledge, it is not known whether this would influence cargo binding (eg Sans).

Response:

We apologize for this mistake. The drawings in all figures were corrected to remove this N-terminal eleven amino-acid extension. The mouse Myo7a isoform used in this study is NM_001256083.1 and it lacks the N-terminal eleven amino-acid extension and 38 amino acids in the third lobe of the first FERM domain.

A recent study reports that this N-terminal extension accelerates ATP hydrolysis and power strokes of MYO7A³. However, we do not expect a significant difference in our conclusion that MYO7A likely traffics as a dimer or an oligomer in stereocilia. It is also unlikely that the MYO7A-SANS interaction is affected since the reported interaction occurs between the CEN domain of SANS and the first lobe of the first MYO7A FERM domain that is present in both isoforms of MYO7A⁴.

Reviewer #2 (Remarks to the Author):

Response:

Thank you very much for co-reviewing our manuscript. We appreciate your contribution.

Reviewer #3 (Remarks to the Author):

General comments:

The authors address an important question in the field: unconventional myosins are poorly understood, particularly in a live-cell context, and the authors correctly point out that traditional single-molecule approaches are not suitable for stereocilia as movement occurs parallel to the plane of imaging. To address this problem, the authors exploit a diSPIM approach, which was established in their previous paper (Miyoshi et al, Cell Rep 2021), to image tagged myosin in live stereocilia. The approach itself is therefore not novel but its use in imaging myosin at the single-molecule level is.

Methods are well reported although the number of biological and technical replicates need to be more clearly stated and there needs to be more analysis of the data (see later).

However, the work itself does not meet the expected standards in the field for 3 primary reasons:

- 1) Data quality is not always clear
 - 2) There is little to no evidence of repeats, with $n = 1$ for the majority of the claims.
 - 3) There are no statistics and therefore no evidence to support any of the conclusions made
- These are major issues that prohibit publication.

Response:

Thank you for your helpful suggestions. All experiments were performed at least three times, which admittedly was not clearly mentioned in the original manuscript. In the revised Methods section, we added the following sentence at the end of the third paragraph in the “Single-molecule microscopy workflow in stereocilia of live hair cells” subsection.

Page 18, Line 563:

“All single-molecule microscopy experiments were performed using at least three different cells from three different sensory epithelia.”

All statistical analysis procedures were described in the revised “Statistical analyses” subsection of the Methods section.

Page 19, Line 593:

“Fluorescent puncta in Fig. 1c were classified using the Gaussian Mixture model⁶⁰ available in the scikit-learn package (<https://scikit-learn.org/>). The rate constant (K) and half-life ($T_{1/2}$) in Fig. S1 were determined by fitting the one-phase decay model implemented in GraphPad Prism. These values were compared against a null hypothesis that one model explains all data using the extra-sum-of-squares F-test implemented in GraphPad Prism¹¹⁹ followed by the correction of P values using the conventional Bonferroni method¹²⁰. The frequency, velocity and run lengths of directionally moving molecules were compared using a two-sided Student’s t -test or one-way ANOVA followed by the post-hoc Tukey test implemented in GraphPad Prism. A directionally moving molecule was defined as a molecule showing a trajectory in the same direction for more than three frames in a kymogram. Molecules moving back and forth in a stereocilium are presumed to be in random molecular motion and were removed from tracking. The frequency of directionally moving molecules was calculated using time-lapse images acquired for more than 500 frames. In Fig. S3a, the ratio of hair cells accumulating HaloTag-fused MYO7A at stereocilia tips was compared using the Pairwise comparisons using Fisher’s exact test implemented in the `fisher.multicomp` function in R (<https://cran.r-project.org/>)”

Specific comments:

Given the lack of repeats or statistics, it is hard to write specific comments. However, here are some (hopefully) constructive feedback for the authors to turn this work into a publication with MAJOR points indicated (the rest of the points are MINOR).

Introduction

- Paragraph 2 on the single-molecule microscopy breaks the flow of the intro. I suggest placing it after introducing the biological system, i.e., after introducing myosin and before the final paragraph of the intro, and use that paragraph to explain why single-molecule imaging is the best way to understand myosin.
- For someone who is not an expert on MET channels, please explain possible ways in which myosin could affect the MET channels before you then show data highlighting that one model is correct - what are the possible ways they could act before you collect your data e.g. they could tether the tip link complex, transport channels to the tip etc.

Response:

Thank you for your suggestions. We reorganized the Introduction section and recorded our edits using Track Changes. The paragraph on single-molecule microscopy was shortened and integrated into the final paragraph as follows:

Page 4, Line 109:

“In this study, we develop a methodology for single-molecule fluorescence microscopy applicable to organelles protruding from the apical surface of tissue and visualize myosin molecules at work in stereocilia (Fig. 1). To make the focal plane coincident with stereocilia, we employ a dual-view inverted selective plane illumination microscope (diSPIM)⁵⁰, which we previously used for multiplexed super-resolution microscopy to detect single molecules of fluorescently labeled imaging probes⁵¹.”

Regarding the contribution of MYO7A to MET channel function, there are several studies of mice with defective MYO7A function that cannot localize crucial components of the tip-link complex and show profound hearing loss due to a lack of MET currents. To clarify, we added the following sentences in the second paragraph of the revised Introduction.

Page 3, Line 74:

“For example, MYO7A interacts through its tail with two scaffolding proteins, SANS and USH1C (harmonin), and localizes them to the UTLD of mature stereocilia¹⁷. UTLDs connect interstereociliary tip links, which consist of CDH23 dimers and PCDH15 dimers^{18,19}, to the F-actin core of longer stereocilium on the CDH23 side²⁰. The PCDH15 side is connected to the MET channel complex which is composed of TMC1/TMC2 channel proteins and accessory proteins, TMIE, CIB2 and LOXHD1²¹⁻²⁵, at the tip of a shorter stereocilium. Among MYO7A’s cargos, harmonin b isoforms harboring the F-actin binding Proline, Serine and Threonine-rich (PST) domain²⁶ are essential for forming the UTLD and tethering tip links to the F-actin core²⁰. Harmonin b and PCDH15 are absent near stereocilia tips in mice with defective MYO7A function (*Myo7a*^{4626SB/4626SB})²⁶⁻²⁸”

Result 1: Single-molecule microscopy in stereocilia of live hair cells

- Fig. 1a Plasmids generated are a lot of work and great to see that it all works. Approach to get plasmids into these cells using the Helios® gene-gun also seemed really successful and cool. Please provide the efficiency of the approach (% of cells labelled) so others can assess using it.

Response:

We agree and have added information on the transfection efficacy. There are approximately ten transfected hair cells in each neonate sensory epithelium. The transfection efficiency is around 0.6% considering that each utricle or saccule has about 1,500 hair cells at birth^{5,6}. However, we do not rely on a high transfection efficiency in these experiments. Rather, it is beneficial to compare hair bundle appearance of transfected and a neighboring non-transfected cells. We assume that users may want to know how many cells maintain intact stereocilia bundles after transfection. To convey this information, we added the following sentence at the end of the first paragraph in the “Single-molecule microscopy workflow in stereocilia of live hair cells” subsection.

Page 5, Line 140:

“In each sensory epithelium explant transfected using plasmid-coated 1.0-µm gold microcarriers, 5–10 hair cells usually express a sufficient amount of HaloTag-fused protein of interest and maintain intact stereocilia architecture.”

- (MAJOR) Fig. 1b In the movies, there are some parts of cells with a lot of over-labelling and it can be very hard to assess if you are truly looking at single molecules or clumps of molecules. This makes it hard to use automated software to pick out the dots and the hand picking of dots is really not ideal. Why not use photoactivatable PA-JF549 to reduce overlap of molecules so you can more sure that you are looking at single molecules and so you can image more molecules per cell without worrying about labelling density? Is it because there is no 405 nm photoactivation laser available?

Response:

We agree with your comment, but there are a few reasons why we did not use photoactivatable dyes. When we developed the methodology, we tested various fluorescent dyes including PA-JF549 and PA-JF646. However, these dyes were not useful for three reasons. First, some of these dyes transition to the bright state without activation by a 405-nm laser. The initial labeling density is uncontrolled because the proportion of dye molecules in the bright state varies even between aliquots of a dye. Second, it is difficult to activate a small number of dye molecules under the diSPIM. Even weak 405-nm illumination activates many dye molecules all over stereocilia and the hair cell body. Finally, and most importantly, these photoactivatable dyes easily transition to the dark state compared with regular dyes. For these reasons, we decided to use non-photoactivatable dyes and instead controlled the label density by optimizing the concentration of dyes.

In the Results section, this finding was mentioned in the second paragraph of the “Single-molecule microscopy workflow in stereocilia of live hair cells” subsection as follows.

Page 6, Line 151:

“Photoactivatable dyes, such as PA-JF549⁵⁹, are not useful for adjusting the labeling density because we found that an uncontrollable proportion of these dyes are in the bright state before being activated by the 405-nm laser and that these dyes are more susceptible to the excitation laser than non-photoactivatable dyes.”

- (MAJOR) Fig. 1c What is the background level when you don't add dye? You can't say puncta were classified into 2 populations because you split the dots in half yourself. You'll need to use a classifier e.g. gaussian mixture model with BIC analysis to show this.

Response:

Fluorescent puncta are not detected without dyes. Mammalian cells usually have a few autofluorescence spots near the nucleus, but they are usually “masked” by the majority of true fluorescent puncta. We agree with the necessity of using a classifier. We analyzed fluorescent puncta including those in Fig. 1b (0.01 nM) using the Gaussian Mixture model provided in the scikit-learn Python library and found that more than 90% of the puncta were classified into two populations. This result is reflected in Fig. 1c with a revised legend as follows.

Page 21, Line 660:

“c, Classification of fluorescent puncta using the Gaussian Mixture model⁶⁰. The sum intensity is calculated by adding all pixel values encompassing each punctum and subtracting the background intensity. Three populations with different peak intensities (Pop1, Pop2 and Pop3) are detected. The peak intensity of Pop2 (985) is approximately

twice as large as that of Pop1 (408) indicating that Pop1 and Pop2 are emitted from one and two fluorophores, respectively. A total of 76 puncta are analyzed from 6 cells including the cell shown in **b**, 0.01 nM, using an average projection of 12 planes for each volume scan.”

We also rephrased a few sentences in the “Single-molecule microscopy workflow in stereocilia of live hair cells” subsection of the Results section.

Page 6, Line 154:

“Single-molecule microscopy is confirmed by calculating the sum of the intensity of each fluorescent punctum and classifying them using a Gaussian Mixture model⁶⁰ (Fig. 1c). Among the three populations detected (Pop1, Pop2 and Pop3), more than 90% of fluorescent puncta are explained by Pop1 and Pop2 (42% and 49%, respectively). The peak intensity of Pop2 (985) is approximately twice as large as that of Pop1 (408) indicating that Pop1 and Pop2 originate from one and two fluorophores, respectively.”

- Fig. 1d You will need to get the PSF of all your molecules and then make error bars or a 95 % confidence interval from the molecules.

Response:

We apologize for the missing error bars. The original Fig. 1d showed the average line intensity profiles of all Pop1 puncta in Fig. 1b (0.01 nM), but error bars were not added. During the revision, we noticed that polynomial interpolation allows for a more precise alignment of line profiles before averaging. We applied this change to Fig. 1d and also showed a smoother PSF of our objective lenses. The Fig. 1d legend now reads:

Page 21, Line 666:

“**d**, Comparison between the average line intensity profile of fluorescent puncta (orange solid line) and the theoretical point spread function (PSF) of the objective lens calculated using the Born & Wolf 3D Optical Model^{117,118} (black dashed line). The similarity between both intensity curves suggests that these puncta are emitted from a point source. Fluorescence intensity is an average of 10 puncta in **b** (0.01 nM, Pop1 only). SD, orange dotted lines.”

We also described the details of the analyses in the revised Methods section. The last paragraph of the “Single-molecule microscopy in live hair cell stereocilia” subsection now includes the following sentences:

Page 18, Line 567:

“Intensity profiles of fluorescent puncta were obtained using the `profile_line` function of `scikit-image` (<https://scikit-image.org/>) and quadratically interpolated using the `interp1d` function of `SciPy` (<https://scipy.org/>). These line profiles were normalized between 0 and 1 and averaged after being aligned to the intensity peaks found by the `find_peaks` function (`SciPy`). The point spread function of the 40x objective lens was calculated using the PSF generator (<https://bigwww.epfl.ch/algorithms/psfgenerator/>) based on the Born & Wolf 3D Optical Model^{117,118} where Refractive index immersion = 1.33, Wavelength = 576 nm⁵⁷ and NA = 0.8.”

- (MAJOR) Authors show single-step photobleaching traces as evidence molecules disappear in a single frame which is good but the structures also move in the videos so a fixed sample will be needed to show this is not movement of the cilia. This will also give the precision of the data and

the photobleaching rate that should be reported in the text and shown as a dotted line when plotting movement parameters.

Response:

We agree with your suggestion. Both fixed and live samples were used to establish our single-molecule microscopy workflow, but the data from fixed samples were not included in the original manuscript. In the revised manuscript, fixed-cell data in Fig. S1b–d show (1) the decay rate of HaloTag-actin puncta during time-lapse imaging, (2) the photobleaching rate of JFX554 dyes and (3) the quantum behavior of JFX554 dyes conjugated to HaloTag-actin molecules. Interestingly, the decay of HaloTag actin occurs more rapidly in live cells than in fixed cells. To describe these data, the last paragraph of the revised “Single-molecule microscopy workflow in stereocilia of live hair cells” subsection now includes the following sentences:

Page 6, Line 171:

“In contrast, most HaloTag-actin molecules stay in one place and show trajectories parallel to the time axis (*X*-axis) of kymograms (Fig. 1f, arrows; Movie S2). In live hair cells, regression of HaloTag-actin molecules is approximately 30-fold slower ($P < 0.001$) than non-fused HaloTag ($T_{1/2} = 13.9$ [13.3–14.5] frames, Fig. S1b) indicating the presence of HaloTag-actin molecules stably bound to the F-actin core. Interestingly, regression of HaloTag-actin molecules occurs more slowly ($P < 0.001$) in fixed cells ($T_{1/2} = 107$ [69–231] frames, Fig. S1b) at a rate close to the “photobleaching” (i.e., decay and transition to the dark state⁶¹) rate of JFX554 dyes ($T_{1/2} = 61$ [52–72] frames, Fig. S1c). HaloTag-actin molecules in live cells may “photobleach” more rapidly or dissociate from actin filaments more easily than in fixed cells. Alternatively, some HaloTag-actin molecules may transiently bind to the F-actin core in live cells⁶². Kymograms show sudden disappearance of HaloTag-actin molecules consistent with the quantum behavior of single fluorophores (Fig. 1f, arrows). The quantum behaviors of dyes are more clearly visualized in control fixed cells acquired every 100 milliseconds (ms) including dyes recovering from the dark state to the bright state (Fig. S1d).”

- (MAJOR) Authors say that, at 1 second time resolution, diffusing molecules are there one frame but actin molecules last ... the video does show this but the conclusion will need $n = 3$ cells at least. It will also need track lengths to be calculated for the various samples (ideally with a fixed cell control) and compared using a stats test, especially as some of the diffusing molecules do get stuck so the relative difference will be important.

Response:

We agree and show track lengths of non-fused HaloTag puncta as a dwell-time distribution in Fig. S1a. To compare track lengths between non-fused HaloTag and HaloTag-actin, we generated a survival curve of non-fused HaloTag puncta from its dwell-time distribution and plotted it with the regression curve of HaloTag-actin puncta in Fig. S1b. We measured regression from $t = 0$ for HaloTag-actin because (1) most HaloTag-actin molecules stably bind to the F-actin core, (2) JFX554 dyes conjugated to HaloTag-actin often “blink” and mimic new binding events of HaloTag-actin, and (3) track lengths (lifetimes) of HaloTag-actin puncta do not always represent the duration of binding to the F-actin core. The data in Fig. S1a is now mentioned in the “Single-molecule microscopy workflow in stereocilia of live hair cells” subsection:

Page 6, Line 166:

“Consistent with diffusion, most non-fused HaloTag molecules stay in the same position no longer than one frame (Fig. 1e, arrows; Movie S1) except for a few molecules that remain in one place probably due to non-specific binding to the stereocilium structure (Fig. 1e, open arrows). This finding is also supported by the short half-life ($T_{1/2}$) of non-fused HaloTag molecules dwelling in stereocilia ($T_{1/2} = 0.46$ [95%CI: 0.45–0.47] frames, Fig. S1a).”

We also apologize for not mentioning the number of replications. All experiments were conducted at least in three different cells from three different sensory epithelia. The number of molecules and cells was mentioned in the legend for all graphs. We also appended the following phrase at the end of the third paragraph in the “Single-molecule microscopy workflow in stereocilia of live hair cells” subsection in the Methods section.

Page 18, Line 563:

“All single-molecule microscopy experiments were performed using at least three different cells from three different sensory epithelia.”

Result 2: Visualization of directional movement of MYO7A dimers in stereocilia
- (MAJOR) Authors say MYO7A shows “continuous trajectories consistent with processive movements” that “move toward the barbed ends of unidirectional F-actin bundles in” but these conclusions cannot be made without more trajectories and statistics. Kymograms and videos show many dots that don’t move at all (Fig.2d-f) so how are molecules chosen to calculate the movement? Are static molecules not shown or are the non-moving dots considered to be background? Surely percentage of non-moving molecules should also be reported. Statistics are also key to report movement e.g. compare molecules +/- AP20187, measure velocity and show statistical changes and report p-values (especially as molecules are sometimes not moving at all). To report directionality, the standard way is to compare the angle from one displacement to the next (or from every n displacements to the next n displacements if movement is slow) and calculate the angle difference e.g. using Circstat on MATLAB and then use a fold anisotropy metric $f(180/0)=P(180^\circ \pm 30^\circ / 0^\circ \pm 30^\circ)$ to define how many-fold more likely a molecule is to make a step backwards compared to a step forward +/- AP20187. You could also define the angle as towards or away from the tip?

Response:

Thank you for your comments. We identified directionally moving molecules using kymograms seeking trajectories for more than three frames in the same direction. Molecules moving back and forth in stereocilia were not included in analyses because they are diffusing molecules. There are two reasons why static molecules were not analyzed. First, we cannot know whether they are bound to stereocilia non-specifically or through their motor domain. Second, the exact number of non-moving molecules cannot be determined because some of the “newly-bound” molecules may be due to the “blinking” of fluorescent dyes (see Fig. S1d). We did not analyze angles since movements of myosin molecules are quasi-one-dimensional in our microscope configuration due to the small (300–500 nm) diameter of vestibular stereocilia⁷. We recognize that analyses of displacements are crucial in myosin studies. The first author has started a research collaboration project to detect myosin molecules in stereocilia at high resolution.

To clarify our criterion in the analyses, we added the following two sentences to the “Statistical analyses” subsection in the revised Methods section:

Page 19, Line 600:

“A directionally moving molecule was defined as a molecule showing a trajectory in the same direction for more than three frames in a kymogram. Molecules moving back and forth in a stereocilium are presumed to be in random molecular motion and were removed from tracking.”

We agree with the importance of statistical analyses and compared the frequency of moving molecules between non-treated and AP20187-treated cells (Fig. S2c). Run lengths are also shown in Fig. 2f. These results are described in the last paragraph of the “MYO7A dimers directionally moving in stereocilia” subsection in the Results section.

Page 7, Line 210:

“The average velocity is 101 ± 53 nm/s ($n = 42$; average \pm SD), which is 10-fold faster than the movements of human recombinant MYO7A-HMM fused with a leucine zipper dimerization sequence on permeabilized filopodia (9.5 ± 0.4 nm/s; average \pm SE)⁶⁷. This difference can be partially attributed to the temperature (37°C in our study vs. 25°C in the previous study) although MYO10 shows only a 2-fold increase of velocity in live-cell filopodia (578 ± 174 nm/s at 25°C vs. 840 ± 210 nm/s at 37°C; average \pm SD)⁶⁸. Run lengths are 2.3 ± 1.0 μ m (Fig. 2f, $n = 42$) and longer than the previous *in vitro* study using single actin filaments (0.71 ± 0.09 μ m; average \pm SE)⁶⁷. In stereocilia of live hair cells, MYO7A-HMM dimers may be allowed to stably bind to F-actin because protein diffusion is restricted by the plasma membrane⁶⁹. HaloTag-MYO7A-HMM-FKBP does not show directional movements without AP20187 (Fig. 2g; Movie S4). However, it is notable that some MYO7A-HMM-FKBP monomers may relocate in stereocilia (Movie S4, circles) and appear as step-wise trajectories in kymograms (Fig. 2g, open arrows in the inset). The frequency of processive movements is semi-quantified assuming that each transfected hair cell expresses an almost equal amount of HaloTag-fused myosin. Processive movements are detected at a significantly higher frequency in AP20187-treated cells than in non-treated cells (Fig. S2c, $P = 0.0004$).”

Result 3: Constitutively active MYO7A mutants move directionally in stereocilia - (MAJOR) Authors say “HaloTag-fused full-length MYO7A did not show directional movements at a detectable frequency in stereocilia”, “Movements of MYO7A-RK/AA were directional and processive resembling the movements of MYO7A-HMM dimers” etc but as above, how many molecules, where are movement and directionality quantification and statistics?

Response:

We apologize for not including quantification and statistical analyses. We compared the frequency of directional movements between full-length MYO7A, MYO7A-RK/AA and MYO7A- Δ SH3- Δ M/F2. The velocity and run lengths were compared between MYO7A-HMM dimers, MYO7A-RK/AA and MYO7A- Δ SH3- Δ M/F2. MYO7A-RK/AA showed directional movements at a significantly higher frequency than full-length MYO7A. However, there was not a significant difference between movement frequencies of MYO7A- Δ SH3- Δ M/F2 and full-length MYO7A, although we did not detect directional movements of full-length MYO7A after analyzing five additional cells. The velocity and run lengths were similar among MYO7A-HMM dimers, MYO7A-RK/AA and MYO7A- Δ SH3- Δ M/F2. These data are shown in Fig. S3, b–d, and mentioned as follows in the last paragraph of the “Processive movements of constitutively active MYO7A mutants in stereocilia” subsection of the Results section.

Page 8, Line 247:

“MYO7A-RK/AA molecules show processive movements more frequently than full-length MYO7A and MYO7A- Δ SH3- Δ M/F2 (Fig. S3b, $P = 0.0010$ and 0.0028), which is consistent with the frequent accumulation of MYO7A-RK/AA at stereocilia tips compared with full-length MYO7A and MYO7A- Δ SH3- Δ M/F2 (Fig. S3a). The velocity during directional movements is not significantly different among MYO7A-RK/AA, MYO7A- Δ SH3- Δ M/F2 and MYO7A-HMM dimers (Fig. S3c). MYO7A-RK/AA and MYO7A- Δ SH3- Δ M/F2 show statistically shorter run lengths than MYO7A-HMM dimers (Fig. S3d, $P = 0.0087$ and 0.037) although this small difference may not be biologically important. Processive movements of MYO7A- Δ SH3- Δ M/F2 indicate that MYO7A can dimerize (or oligomerize) using motifs in the neck or the first MyTH4-FERM (M/F1) domain. The SH3 and/or M/F2 domains may be necessary for MYO7A to traffic efficiently in stereocilia., For example, harmonin can interact with the M/F2 domain and expose the motor domain which can lead to dimerization of MYO7A⁷³.”

Result 4: Distinct behavior of MYO7A and MYO10 anchored to the plasma membrane - (MAJOR) Authors again make a series of conclusions without stats e.g. “Movements of membrane-anchored MYO7A-HMM were different from MYO7A-HMM dimers or constitutively active MYO7A mutants”, “Directional movements of MYO10-MD indicate that the restricted movements of MYO7A-HMM are derived from the kinetic differences between the motor domains of MYO7A and MYO10” (are they? What is the p-value?) They do quantify speed this time and show stdev which means difference is significant (although it is best to report the p-value), which is great, but length of track is also an important parameter. It looks like the AP21987 slows it down so then it takes longer to get to the tip i.e. slower and longer tracks which lead to the same total distance covered.

Response:

We agreed and compared the frequency of directional movements between (1) unanchored MYO7A-HMM and membrane-anchored MYO7A-HMM, and (2) unanchored MYO10-MD and membrane-anchored MYO10-MD. The frequency of directionally moving molecules was not compared between MYO7A-HMM and MYO10-MD because MYO7A-HMM's step-wise movements are qualitatively different from MYO10-MD's processive movements. Velocity and run lengths were compared between membrane-anchored MYO7A-HMM, unanchored MYO10-MD and membrane-anchored MYO10-MD using the MYO7A-HMM dimer as a control. Movements of membrane-anchored MYO7A-HMM were significantly slower than both unanchored and membrane-anchored MYO10-MD. Membrane-anchored MYO7A-HMM showed slightly shorter run lengths than MYO7A-HMM-dimers and membrane-anchored MYO10-MD. The data are summarized in Fig. S4 and described in the last paragraph in the “Membrane-anchored MYO7A does not show processive movements” subsection of the Results section.

Page 9, Line 281:

“Single-molecule microscopy shows that MYO7A anchored to the plasma membrane does not show processive movements in stereocilia. Instead, we find that membrane-anchored MYO7A-HMM sometimes shows step-wise directional movements (Fig. 4d; Movie S7). However, step-wise movements may not be solely due to anchoring to the membrane because MYO7A-HMM monomers show similar movements at a low frequency (Fig. 2g). Membrane-anchored MYO7A-HMM shows step-wise movements at a higher frequency than unanchored MYO7A-HMM, but this difference is not statistically significant (Fig. S4a). In contrast, MYO10-MD markedly changes its movements by membrane anchoring (Fig. 4e; Movies S8 and S9) indicating that IL2R α -EGFP-FKBP does function as a scaffold for some myosins to traffic in stereocilia. Consistently, with its

weak accumulation at stereocilia tips (Fig. 4c, arrowhead) and filopodia⁵², MYO10-MD can show rapid directional and processive movements toward stereocilia tips without anchoring to the membrane (Fig. 4e, arrows and open arrows in the upper panel). After AP21987 treatment, MYO10-MD begins to show slow processive movement toward stereocilia tips (Fig. 4e, arrows and open arrows in the lower panel, also kymograms in Fig. S4c) and sometimes processive retrograde movement (Fig. S4c, arrowheads). Although the frequency of processive movements does not change significantly by membrane anchoring (Fig. S4b), the velocity of processive movements is significantly different between MYO10-MD monomers ($1,800 \pm 490$ nm/s, $n = 14$) and membrane-anchored MYO10-MD (701 ± 297 nm/s, $n = 32$) (Fig. S4d, $P < 0.0001$). Membrane-anchored MYO7A-HMM moves at 88 ± 27 nm/s ($n = 7$), which is not significantly different from MYO7A-HMM dimers (101 ± 53 nm/s) (Fig. S4d) but much slower than MYO10-MD (Fig. S4d, $P < 0.0001$ vs. MYO10-MD monomers and vs. membrane-anchored MYO10-MD). Run lengths are slightly shorter for membrane-anchored MYO7A-HMM compared with membrane-anchored MYO10-MD ($P = 0.018$) and MYO7A-HMM dimers ($P = 0.011$) (Fig. S4e).”

- Fig. 4e shows, as the authors explain, that molecules can move both to the tip and back so quantification of direction will be important here and it would be interesting to compare MYO7A and MYO10 +/- AP21987 using a directional metric to show if it goes from random diffusion to directional motion ... after all, we don't see as many tracks in MYO7A that go away from the tip i.e. it could be more directional.

Response:

We agree with your suggestion and recognize the importance of tracking each molecule with precision. However, approaching this question requires some technical breakthroughs and is beyond the scope of the current study. The first author's laboratory has just started collaborative projects to enable such precise tracking of myosin molecules and to analyze step sizes and directions of their movements.

Result 5: Step-wise movements of MYO7A and MYO10 when tethered to F-actin

- “Movements of MYO7A-HMM molecules tethered to F-actin were step-wise as observed for those anchored to the plasma membrane. MYO10-MD molecules also showed step-wise movements after the AP21987 treatment” Again this needs to be quantified e.g. number of consecutive movements in the same direction that are above the precision limit i.e. that are not static. A histogram of this number of consecutive movements will show whether the steps are always the same size and quantify the distance taken per step. This histogram can then be compared to the previous datasets that are not stepwise to make the point that some traces are more stepwise than others

- “These observations are not consistent with an “inchworm-like” movement proposed by others because the step-sizes were 100–200 nm” – Could there be inchworm-like movement occurring at faster timescales than the time resolution of the videos?

Response:

Thank you for your comments. First of all, we apologize for our misinterpreting the movements of MYO10-MD tethered to F-actin. While we were revising this manuscript, we discovered that MYO10-MD molecules show processive and directional movements

toward stereocilia tips when coupled with a harmonin b fragment containing the PST domain, which is referred to as the DFRCR fragment in the revised manuscript.

We updated Figs. 5 and S5 to include this finding and necessary statistical analyses. In the Results section, the last paragraph of the subsection is now titled “Possible step-wise movements of MYO7A coupled with a harmonin b fragment”. We state:

Page 11, Line 344:

“Only step-wise movements are detected for MYO7A-HMM coupled with the DFRCR fragment (Fig. 5e, arrows; Movie S10). Different from the membrane anchor IL2R α -EGFP-FKBP, the DFRCR fragment may partly function as a scaffold for MYO7A-HMM to move in stereocilia because the frequency of directional movements increases slightly after AP21987 treatment (Fig. S5a, $P = 0.033$). This finding is consistent with the AP21987-dependent accumulation of HaloTag-MYO7A-HMM-FKBP without FRB-DFRCR-EGFP puncta (Fig. 5c, yellow arrows). The CC1 and CC2 domains in the DFRCR fragment are unlikely to dimerize MYO7A-HMM since no processive movements are detected. As shown for IL2R α -EGFP-FKBP, MYO10-MD’s movements slow down when coupled with the DFRCR fragment (Fig. 5f; Movie S11) but without changing the frequency of moving molecules (Fig. S5b). MYO10-MD moves at 831 ± 456 nm/s ($n = 14$) in cells treated with AP21987, which is significantly slower than $1,780 \pm 487$ nm/s ($n = 14$) in untreated cells (Fig. S5c, $P < 0.0001$) suggesting that FRB-DFRCR-EGFP can function as a scaffold for MYO10-MD to move in stereocilia. The velocity is not significantly different between processive movements of MYO7A-HMM dimers (101 ± 53 nm/s) and step-wise movements of MYO7A-HMM coupled with the DFRCR fragment (84 ± 44 nm/s, $n = 12$). MYO7A-HMM moves much slower than MYO10-MD (Fig. S5c). Run lengths of MYO7A-HMM dimers are slightly longer than MYO7A-HMM coupled with the DFRCR fragment, uncoupled MYO10-MD and MYO10-MD coupled with the DFRCR fragment (Fig. S5d, $P = 0.006$, 0.0019 and 0.0024 , respectively). Along with the absent processive movements of membrane-anchored MYO7A-HMM, these results suggest that processive movements of constitutively active mutants used in this study, MYO7A-RK/AA and MYO7A- Δ SH3- Δ M/F2, are driven by their “walk” as a dimer or an oligomer on the unidirectionally bundled actin filaments in stereocilia.”

Reviewer #4 (Remarks to the Author):

In the present manuscript, the authors utilized live-cell single-molecule fluorescence microscopy to examine MYO7A transport in the stereocilia of inner ear hair cells. Using various MYO7A truncations and mutants, they showed that MYO7A traffics in the stereocilia as dimers (or oligomers), and its movement is restricted when scaffolded by the plasma membrane or stereociliary F-actin core. In general, it is an interesting and well-performed work, and is of significance to the related field. Similar strategies could be used to explore the transport of other important proteins in the stereocilia. This reviewer has some concerns and suggestions that are listed below.

1. The data in the present manuscript did not answer how native MYO7A is transported to the stereocilia tips. Is native, full-length MYO7A transported when attached to plasma membrane? Coupled to the F-actin core? Or both? The authors showed that “Halo Tag-fused full-length MYO7A did not show directional movements at a detectable frequency in stereocilia of vestibular hair cells”, and explained that it might because “full-length MYO7A takes a backfolded autoinhibitory conformation between the tail and motor domains” (line 229-232). It’s interesting why endogenous MYO7A can release this autoinhibition in some way, and somehow Halo Tag-

fused full-length MYO7A cannot. Nevertheless, the authors showed that MYO7A RK/AA mutant can move directionally in the stereocilia. What is the velocity of RK/AA mutant movement? Could the behavior of MYO7A RK/AA mutant represent that of the native MYO7A?

Response:

Thank you for your comments and questions. Our current data suggest that (1) MYO7A dimers can move in stereocilia over long distances and that (2) MYO7A can move less frequently and over short distances when tethered to the plasma membrane or F-actin. Combined with the previously reported cargo-mediated activation by MyRIP⁸ and the directional movements of RK/AA and Δ SH3- Δ M/F2, we speculate that some endogenous factors, such as components of the tip-link complex, unleash MYO7A from an autoinhibitory state and mediate dimerization (oligomerization) of MYO7A molecules. We consider that MYO7A RK/AA represents how native untaged MYO7A behaves when unleashed from the autoinhibitory state.

In the revised manuscript, we compared the velocity and run length of RK/AA and Δ SH3- Δ M/F2 with MYO7A-HMM dimers and found that there is little difference among these three MYO7A mutants/fragments. We additionally imaged five hair cells expressing HaloTag-fused full-length MYO7A but did not detect directional movements in these cells. The results are shown in Fig. S3, c and d, with the following phrases in the last paragraph of the “Processive movements of constitutively active MYO7A mutants in stereocilia” subsection in the Results section.

Page 8, Line 247:

“MYO7A-RK/AA molecules show processive movements more frequently than full-length MYO7A and MYO7A- Δ SH3- Δ M/F2 (Fig. S3b, $P = 0.0010$ and 0.0028), which is consistent with the frequent accumulation of MYO7A-RK/AA at stereocilia tips compared with full-length MYO7A and MYO7A- Δ SH3- Δ M/F2 (Fig. S3a). The velocity during directional movements is not significantly different among MYO7A-RK/AA, MYO7A- Δ SH3- Δ M/F2 and MYO7A-HMM dimers (Fig. S3c). MYO7A-RK/AA and MYO7A- Δ SH3- Δ M/F2 show statistically shorter run lengths than MYO7A-HMM dimers (Fig. S3d, $P = 0.0087$ and 0.037) although this small difference may not be biologically important. Processive movements of MYO7A- Δ SH3- Δ M/F2 indicate that MYO7A can dimerize (or oligomerize) using motifs in the neck or the first MyTH4-FERM (M/F1) domain. The SH3 and/or M/F2 domains may be necessary for MYO7A to traffic efficiently in stereocilia., For example, harmonin can interact with the M/F2 domain and expose the motor domain which can lead to dimerization of MYO7A⁷³.”

2. Shin and colleagues reported that there are various MYO7A isoforms with different N-termini (MYO7A-S and MYO7A-C, Li et al, NC 2020). Do these different N-termini affect the transport of MYO7A to the tips of stereocilia?

Response:

A recent study reported that the N-terminal extension accelerates ATP hydrolysis and the power stroke of MYO7A³. Thus, one would expect that this N-terminal extension could affect MYO7A’s trafficking and/or its final localization in stereocilia. However, it is difficult in this study to detect how the N-terminal extension affects the motor activity of MYO7A because HaloTag is fused to the N-terminus of MYO7A to analyze which endogenous protein partner(s) can activate MYO7A’s trafficking in stereocilia and how activated MYO7A molecules traffic under physiological conditions. The N-terminal extension cannot accelerate ATP hydrolysis of the motor domain when GFP is added to

the N-terminus³. Once we understand how MYO7A's motor activity is regulated in stereocilia, we should be able to experimentally address your question by fusing HaloTag to the C-terminus and activating movements of MYO7A containing N-terminal extension with a physiologically relevant method.

3. Several deafness-mutations in the motor region of MYO7A have been identified. Does any of these mutations affect the transport of MYO7A to the stereocilia tips?

Response:

Yes. Approximately 200 different variants in the motor domain of MYO7A are curated in ClinVar (<https://www.ncbi.nlm.nih.gov/clinvar/>) as pathogenic or likely pathogenic variants associated with human hereditary hearing loss^{9,10}. Among these variants, nonsense and frameshift mutations will obviously affect the transport of MYO7A toward stereocilia tips. It is also known that missense mutations affecting the residues facing the cleft in the motor domain, which is an interface with F-actin and an active site for ATP hydrolysis, often cause human (and mouse) hearing loss¹¹. Profound hearing loss in mouse models disabling the MYO7A motor function, such as *Myo7a*^{sh1/sh1} and *Myo7a*^{6J/6J}, also indicates that the motor function of MYO7A is essential for developing functional stereocilia.

4. The models presented in Figure 6c-e are nice. Could the authors clearly state which model their data support the most?

Response:

Thank you for your comments. To clarify, we started the third paragraph of the revised Discussion section with the following sentences below. According to Reviewer 1's suggestion, the Discussion section was reorganized. All edits are visible with Track Changes.

Page 13, Line 408:

"Using the current data, we discuss how tip-link components localize in stereocilia. We consider one unlikely "scaffold first" scenario (Fig. 6d) and two more likely "travel and connect" and "walking links" scenarios (Fig. 6, e and f)."

References:

- 1 Merritt, R. C. *et al.* Myosin IIIB uses an actin-binding motif in its espin-1 cargo to reach the tips of actin protrusions. *Curr Biol* **22**, 320-325 (2012). <https://doi.org/10.1016/j.cub.2011.12.053>
- 2 Houdusse, A. & Titus, M. A. The many roles of myosins in filopodia, microvilli and stereocilia. *Curr Biol* **31**, R586-R602 (2021). <https://doi.org/10.1016/j.cub.2021.04.005>
- 3 Hollo, A. *et al.* Molecular regulatory mechanism of human myosin-7a. *The Journal of biological chemistry* **299**, 105243 (2023). <https://doi.org/10.1016/j.jbc.2023.105243>
- 4 Wu, L., Pan, L., Wei, Z. & Zhang, M. Structure of MyTH4-FERM domains in myosin VIIa tail bound to cargo. *Science* **331**, 757-760 (2011). <https://doi.org/10.1126/science.1198848>
- 5 Li, A., Xue, J. & Peterson, E. H. Architecture of the mouse utricle: macular organization and hair bundle heights. *Journal of neurophysiology* **99**, 718-733 (2008). <https://doi.org/10.1152/jn.00831.2007>
- 6 Miwa, T., Ito, N. & Ohta, K. Tsukushi is essential for the formation of the posterior semicircular canal that detects gait performance. *Journal of Cell Communication and Signaling* **15**, 581-594 (2021). <https://doi.org/10.1007/s12079-021-00627-1>

- 7 Morita, I., Komatsuzaki, A. & Tatsuoka, H. The morphological differences of stereocilia and cuticular plates between type-I and type-II hair cells of human vestibular sensory epithelia. *ORL J Otorhinolaryngol Relat Spec* **59**, 193-197 (1997). <https://doi.org/10.1159/000276939>
- 8 Sakai, T., Umeki, N., Ikebe, R. & Ikebe, M. Cargo binding activates myosin VIIA motor function in cells. *Proceedings of the National Academy of Sciences of the United States of America* **108**, 7028-7033 (2011). <https://doi.org/10.1073/pnas.1009188108>
- 9 Riazuddin, S. *et al.* Mutation spectrum of MYO7A and evaluation of a novel nonsyndromic deafness DFNB2 allele with residual function. *Hum Mutat* **29**, 502-511 (2008). <https://doi.org/10.1002/humu.20677>
- 10 Miyoshi, T., Belyantseva, I. A., Sajeevadathan, M. & Friedman, T. B. Pathophysiology of human hearing loss associated with variants in myosins. *Front Physiol* **15**, 1374901 (2024). <https://doi.org/10.3389/fphys.2024.1374901>
- 11 Sellers, J. R. Myosins: a diverse superfamily. *Biochimica et biophysica acta* **1496**, 3-22 (2000). [https://doi.org/10.1016/s0167-4889\(00\)00005-7](https://doi.org/10.1016/s0167-4889(00)00005-7)

June 24, 2025

Dear Reviewers,

We sincerely thank you for your valuable comments. Please find below our point-by-point responses. All edits of the manuscript are visible in Track Changes.

As part of this revision, we refined the manuscript title and the nomenclature of our experimental workflow to “**Single-molecule fluorescence microscopy reveals regulatory mechanisms of MYO7A-driven cargo transport in stereocilia of live inner ear hair cells**” and “**STELLA-SPIM**”, respectively. We also acknowledged Erich Boger, Phil Jensik and Steve Saltekoff for their contributions. Grammatical and minor logical errors have been corrected and are indicated in Track Changes.

Reviewer #1 (Remarks to the Author):

I thank the authors for addressing my comments in their rebuttal. The modifications made to the revised manuscript have helped strengthen their conclusions.

Response:

We appreciate your valuable comments and suggestions. We are encouraged by your positive feedback and hope that our approach will be useful for researchers in hearing and related fields.

Reviewer #2 (Remarks to the Author):

Response:

Thank you very much for co-reviewing our manuscript again. We appreciate your contribution to the review process.

Reviewer #3 (Remarks to the Author):

We thank the authors for diligently going through and addressing all the reviewer's comments. The paper is now robust and we have no further comments.

Response:

Thank you very much for your kind words. We are pleased that our revised manuscript has met your expectations. We sincerely appreciate your thoughtful suggestions.

Reviewer #4 (Remarks to the Author):

The revised manuscript is greatly improved. The authors addressed most of the questions. However, it's still not clear to this reviewer why Halo Tag-fused full-length MYO7A cannot show directional movements at a detectable frequency in stereocilia. The authors speculate that “some endogenous factors, such as components of the tip-link complex, unleash MYO7A from an autoinhibitory state and mediate dimerization (oligomerization) of MYO7A molecules.” How

does this apply only to endogenous full-length MYO7A, but not Halo Tag-fused full-length MYO7A? Could the authors elaborate more on this?

Response:

Thank you for your insightful comments. We considered two possible explanations for why ectopically expressed, HaloTag-fused full-length mouse MYO7A does not show directional movements. First, the HaloTag may enhance autoinhibition, thereby suppressing MYO7A motor activity. Second, MYO7A may be activated only when its effector proteins, such as SANS and harmonin, are expressed at sufficient levels in hair cell cytoplasm. We experimentally tested the first possible explanation and added new data to this manuscript. The second possible explanation is discussed in the original manuscript, at the end of the second paragraph in the Discussion section.

To examine the first explanation that the HaloTag downregulates MYO7A motor activity, we replaced the large N-terminal HaloTag of full-length mouse MYO7A with a small HA tag and expressed the constructs in vestibular hair cells. The following sentences were added at the end of the first paragraph in the “Processive movements of constitutively active MYO7A mutants in stereocilia” subsection of the Results section, in the “Fluorescence histochemistry” subsection of the Methods section and in the Fig. S3b legend.

Page 8, Line 244:

“It is also notable that HaloTag-fused mouse MYO7A used in this study localizes differently from endogenous MYO7A in guinea pig and rat inner ear, which concentrates at the UTLD¹⁷. To examine the possibility that a HaloTag stabilizes the autoinhibitory conformation of mouse MYO7A and accounts for this discrepancy, we expressed full-length mouse MYO7A in vestibular hair cells after fusing the small HA tag at the N- or at the C-terminus (Fig. S3b). Immunostaining revealed that HA-tagged MYO7A is diffusely distributed along stereocilia, similar to HaloTag-fused MYO7A, suggesting that ectopically expressed MYO7A remains autoinhibited regardless of tag type or position. Additional factors are likely required to activate ectopically expressed mouse MYO7A in mouse hair cells.”

Page 16 Line 515:

“Plasmids pHA-C1 and pHA-N3 to express HA-tagged full-length MYO7A were constructed from pEGFP-C1 and pEGFP-N3 by replacing the EGFP coding sequence with oligonucleotide linkers encoding YPYDVPDYA. The first methionine was added to the HA-tag sequence of pHA-C1 to encode MYPYDVPDYA.”

Page 19 Line 603:

“For immunohistochemistry of the HA tag, samples were blocked and permeabilized in PBS-BSA-Tx for 1 h at RT and then incubated with HA-Tag (C29F4) Rabbit mAb #3724 (Cell Signaling Technology) diluted 1:500 in PBS-Tx for 2 h at RT. After washing in PBS for 5 min three times, the anti-HA tag antibody was visualized using an Alexa Fluor 568-conjugated goat anti-rabbit IgG (H+L) antibody (A-11036, Thermo Fisher Scientific) diluted 1:500 in PBS-Tx for 1 h at RT. F-actin was counterstained with 10–30 nM Alexa Fluor™ Plus 405 Phalloidin.”

Page 29 Line 887:

“**b**, Distribution of ectopically expressed full-length MYO7A, with an HA tag at the N-terminus (HA-MYO7A) or at the C-terminus (MYO7A-HA). Immunostaining for the HA-

tag reveals that full-length MYO7A is diffusely distributed along stereocilia (arrows), similar to the distribution observed with HaloTag-fused full-length MYO7A. Signals from HA-MYO7A or MYO7A-HA are more intense in the cell body (arrowheads). Bar, 5 μ m.”

In page 14, line 427-428, “Fig. 6d” should be “Fig. 6e”; in line 429, “Fig. 6e” should be “Fig. 6f”.

Response:

We apologize for this oversight. The figure references were corrected.